# Characterization of Human Induced Pluripotent Stem Cell-Derived Hepatocytes with Mature Features and Potential for Modeling Metabolic Diseases

**DOI:** 10.3390/ijms21020469

**Published:** 2020-01-11

**Authors:** Gustav Holmgren, Benjamin Ulfenborg, Annika Asplund, Karin Toet, Christian X Andersson, Ann Hammarstedt, Roeland Hanemaaijer, Barbara Küppers-Munther, Jane Synnergren

**Affiliations:** 1Systems biology research center, School of Bioscience, University of Skövde, 54128 Skövde, Sweden; gustav.holmgren@his.se (G.H.); jane.synnergren@his.se (J.S.); 2R&D, Hepatocyte Product Development, Takara Bio Europe AB, 41346 Gothenburg, Sweden; annika_asplund@takarabio.com (A.A.);; 3Department of Metabolic Health Research, TNO, 2333 Leiden, The Netherlands; karin.toet@tno.nl (K.T.); roeland.hanemaaijer@tno.nl (R.H.); 4The Lundberg Laboratory for Diabetes Research, Departments of Molecular and Clinical Medicine, Institute of Medicine, Sahlgrenska Academy, University of Gothenburg, 41345 Gothenburg, Sweden; ann.hammarstedt@astrazeneca.com

**Keywords:** human induced pluripotent stem cells, human stem cell-derived hepatocytes, in vitro, metabolic diseases, transcriptomics, maturation, characterization

## Abstract

There is a strong anticipated future for human induced pluripotent stem cell-derived hepatocytes (hiPS-HEP), but so far, their use has been limited due to insufficient functionality. We investigated the potential of hiPS-HEP as an in vitro model for metabolic diseases by combining transcriptomics with multiple functional assays. The transcriptomics analysis revealed that 86% of the genes were expressed at similar levels in hiPS-HEP as in human primary hepatocytes (hphep). Adult characteristics of the hiPS-HEP were confirmed by the presence of important hepatocyte features, e.g., Albumin secretion and expression of major drug metabolizing genes. Normal energy metabolism is crucial for modeling metabolic diseases, and both transcriptomics data and functional assays showed that hiPS-HEP were similar to hphep regarding uptake of glucose, low-density lipoproteins (LDL), and fatty acids. Importantly, the inflammatory state of the hiPS-HEP was low under standard conditions, but in response to lipid accumulation and ER stress the inflammation marker tumor necrosis factor α (TNFα) was upregulated. Furthermore, hiPS-HEP could be co-cultured with primary hepatic stellate cells both in 2D and in 3D spheroids, paving the way for using these co-cultures for modeling non-alcoholic steatohepatitis (NASH). Taken together, hiPS-HEP have the potential to serve as an in vitro model for metabolic diseases. Furthermore, differently expressed genes identified in this study can serve as targets for future improvements of the hiPS-HEP.

## 1. Introduction

The liver is an organ that spans a large variety of different functions in the body: energy metabolism, detoxification, and production of serum proteins and bile, just to mention a few [1]. Consequently, liver disease or liver toxicity that cause impaired liver functionality have severe effects on normal body functions. Liver diseases are a major burden to the public health and estimations show a global liver-related mortality rate of approx. 2 million patients per year [2]. In addition, the liver is one of the organs with the highest susceptibility to drug toxicity, significantly contributing to the high attrition rates in current drug discovery processes [3]. Therefore, there is a strong need for better and more predictive in vitro models for liver disease and toxicity. A general opinion within the field of safety assessment and disease modeling is that in vitro cell models will increasingly contribute to improve the mechanistic understanding of diseases and the prediction of adverse effects of drugs in humans in the future.

The gold standard for studying liver disease or hepatotoxicity in vitro are metabolically competent human primary hepatocytes (hphep), either freshly isolated or cryopreserved. However, the hphep have several shortcomings that limit their use. The shortage of relevant human donor material, the large donor variation, and the rapid loss of their functionality, e.g., of the expression of the drug metabolizing machinery, in culture [4,5,6,7], are the most prominent problems. Furthermore, the liver consists of multiple cell types; hepatocytes and non-parenchymal cells, such as cholangiocytes, endothelial cells, hepatic stellate cells, and Kupffer cells. As a consequence, an in vitro liver model with capacity to display all hepatic functions needs to consist of several if not all hepatic cell types and not only hepatocytes. In addition to using multiple cell types, a 3-dimensional (3D) culture environment is beneficial for an in vitro model as it allows the formation of cell-cell contacts and the formation of cell polarity in hepatocytes as indicated by the presence of bile canaliculi [8].

Nonetheless, functional hepatocytes will constitute the major part of a relevant in vitro liver model, and therefore, stem cell research has been focused on deriving mature hepatocytes from stem cells. Due to the many diverse features of hepatocytes, and also considering the fact that primary hepatocytes rapidly lose key functions when cultured in conventional 2D cultures in vitro, it is not surprising that hepatocytes are one of the most challenging cell type to derive from stem cells and further mature in vitro.

In order to explore the potential of a cell model and identify good-to-go areas for the cell model, a thorough characterization of the cells in question is necessary. This characterization needs to be done on multiple levels, preferably combining large-scale transcriptomics assessment with multiple functional assays [9]. Importantly, hepatocytes exist in periportal and perivenous phenotypes in the liver lobe, commonly known as metabolic zonation [10], and it is important to look at features for both these hepatocyte populations when assessing a hepatocyte cell model.

Human induced pluripotent stem cells (hiPSC) are a virtually unlimited source of cells and have the potential to differentiate into specialized cell types, which provides unique opportunities for usage in a wide range of applications. The possibility to convert hiPSCs into functional hepatocytes allows for novel opportunities in assay development and holds great potential for future pre-clinical and regenerative medicine applications. For example, hiPSC derived from patients suffering from, e.g., inherited metabolic diseases can be used for modeling these types of diseases in vitro [11,12,13]. Combining the hiPSC technology with the constantly progressing genome editing technologies, e.g., CRISPR-Cas9, further increases the potential for disease modeling and therapeutic applications [14,15].

The study presented here describes a thorough characterization of hiPSC-derived hepatocytes (hiPS-HEP), in order to assess their potential as in vitro tools for metabolism studies and disease modeling. The cells were subjected to a broad panel of characterization assays, ranging from transcriptomics analysis, protein expression, to multiple functional assays. The results show that the hiPS-HEP possess many adult hepatocyte features that can be maintained in conventional 2D cultures for over 2 weeks. Furthermore, primary hepatic stellate cells could be activated in 2D co-cultures with hiPS-HEP which, to the best of our knowledge, is the first time that this is reported. In addition, 3D co-culture spheroids were established, paving the way for using these for modeling non-alcoholic steatohepatitis (NASH). Importantly, the transcriptomics analysis revealed that a majority of genes (86%) are similarly expressed in all the three test groups. Taken together, this study highlights the potential of hiPSC-derived hepatocytes as a robust and reproducible source for hepatocyte-related in vitro models, e.g., metabolism-related liver diseases.

## 2. Results

In the present study, the maturation status and the functionality of the hiPS-HEP was compared to hphep by combining large-scale transcriptomics with multiple functional readouts covering a variety of important hepatocyte functions, as suggested by Schwartz et al. [9] amongst others. In order to assess the reproducibility and robustness of the used differentiation protocol, two batches of hiPS-HEP derived from three different hiPSC lines were tested and compared to hphep from 3 different donors.

### 2.1. HiPSC-Derived Hepatocytes with Highly Uniform HNF4α Expression and Mature Hepatocyte Features

The hiPS-HEP cultures consisted of large cells with a polygonal cell shape and a dark cytoplasm (Figure 1C1), which is typical for hepatocytes. HiPS-HEP were plated at confluency after thawing and did not proliferate, indicating terminal differentiation. Importantly, expression of stemness markers such as Oct4 (*POUF51*) were as low in hiPS-HEP as in hphep (see heatmap in Figure 1B), which indicates the loss of the stemness features as a result of an efficient hepatocyte differentiation. For a first general assessment, the expression of typical hepatocyte-related markers such as albumin (*ALB*), asialoglycoprotein receptor 1 (*ASGPR1*), connexin 32 (*GJB1*), hepatocyte nuclear factor 4α (*HNF4α*), and α1-antitrypsin (*AAT*) were investigated in hiPS-HEP and these markers were found to be expressed at similar mRNA levels as in hphep (Figure 1A). As expected, the mRNA expression of these five markers decreased in the hphep after culturing for 24 h (hphep d0 vs. d1). Next, protein expression of these hepatocyte markers was investigated using immunocytochemistry (ICC; Figure 1C,D). Almost all cells in the hiPS-HEP and hphep cultures were immuno-positive for HNF4α (Figure 1C2 and C6, respectively; Appendix A). Quantification using CellC Cell Counting software showed that more than 92% of the cells in hiPS-HEP cultures derived from the three hiPSC lines ChiPSC12, ChiPSC18 and ChiPSC22 expressed HNF4α, counted in relation to 4′,6-diamidino-2-phenylindole (DAPI) staining (Appendix A). The few HNF4α-negative cells are also likely to be derived from definitive endoderm (DE) since these cultures are near-homogenous with 97% Sox17-immunopositive cells [16].

In contrast to the highly uniform HNF4α expression, ASGPR1, α1-Antitrypsin (AAT), and Albumin (Alb) appeared to be more strongly expressed in a subpopulation in both hiPS-HEP and hphep cultures (Figure 1C3–C5 and C7–C9, respectively). The immunostaining of these three markers was comparable for hiPS-HEP derived from all three hiPSC lines in terms of staining intensity (Appendix A).

In agreement with *Albumin* mRNA expression levels and immunostainings, Albumin secretion was found to be at comparable levels in hiPS-HEP and hphep (Figure 1D). Importantly, Albumin secretion by hiPS-HEP was stable for over two weeks in culture (between day 4 and 20 post-thaw) indicating phenotypical stability. In order to evaluate urea secretion, another important liver specific function, the mRNA expression levels of key enzymes of the urea cycle and other related genes were compared in hphep and hiPS-HEP. Several urea cycle-related genes were expressed at comparable mRNA levels in hiPS-HEP and hphep (Figure 1B), whereas *ARG1*, *ASL, ASS1, CPS1, OTC,* and *SLC25A2* were expressed at significantly lower levels in hiPS-HEP compared to hphep directly after thawing (d0). When measuring urea secretion upon ammonium challenge, hiPS-HEP displayed lower urea secretion than hphep cultured for 1 day post-thaw (d1; Figure 1E) which may be due to the lower expression of *ARG1* in hiPS-HEP compared to hphep d1 (Figure 1B). Similarly, to the stable Albumin secretion, urea secretion was stable in hiPS-HEP between day 13 and 20 post-thaw (Figure 1E).

### 2.2. Expression of Drug Metabolizing Enzymes and Transporters

Expression and functionality of the drug metabolizing machinery, comprising phase I and phase II enzymes as well as transporter proteins, is of critical importance for the utility of an in vitro hepatocyte model in drug metabolism and hepatotoxicity studies but of less importance for disease modeling. Since several enzymes are known to be specific for fetal or adult hepatocytes, analyses of these enzymes can also aid to assess the grade of maturity of the cells.

We started by investigating the most important Cytochrome P450 (CYP) enzymes and found that several CYP enzymes were expressed in hiPS-HEP at similar mRNA levels as in hphep d1, e.g., *CYP2C19, 2C9, 3A4, 3A5*, and *3A7* (Figure 2A). In contrast to that, other enzymes such as *CYP1A2, CYP2B6,* and *CYP2D6*, were expressed at lower mRNA levels in hiPS-HEP compared to hphep d1 (Figure 2A). Importantly, the similar mRNA levels of the adult enzyme *CYP3A4* and the fetal enzyme *CYP3A7* in hiPS-HEP and hphep indicate an adult feature of hiPS-HEP. Since a low correlation between mRNA and activity levels for many phase I and II enzymes as well as transporters has been reported by Ohtsuki et al. [17], we also assessed the enzyme activity levels by incubating with specific substrates for the different enzymes and measuring the formation of the specific metabolites with LC/MS. We found that hiPS-HEP had similar CYP1A, CYP3A, and CYP2C9 activities as hphep cultured for 20 h (including the activity assay; Figure 2D1–D3). In contrast, CYP2B6 and CYP2D6 activity levels were lower in hiPS-HEP compared to hphep (Figure 2D) which is in agreement with the mRNA expression (Figure 2A). Notably, CYP activities in hiPS-HEP were stable or even increasing for a period of 14 days, between day 4 and 19 (Figure 2D1–D6). A discrepancy between CYP2C19 and CYP2C9 mRNA and activity levels could be observed. *CYP2C19* mRNA expression is on the same level in the two cell types, whereas the activity is approximately 10 times lower in hiPS-HEP than in hphep (Figure 2A,D6). However, the LC/MS method does not seem to be as stable for detection of the CYP2C19 metabolite as for the other metabolites. Regarding CYP2C9, lower mRNA levels were observed in hiPS-HEP than in hphep but the activity levels were similar in both cell models (Figure 2A,D3). The discrepancies seen between mRNA and activity levels for some CYP enzymes are not unexpected. Ohtsuki et al. have reported previously that CYP2C9 and CYP2C19 are among the CYP enzymes with the lowest correlation between mRNA and activity levels [17]. This clearly emphasizes that protein expression or preferably functional assays are essential when characterizing in vitro hepatocyte models.

Since drug metabolism is also performed by phase II enzymes, we investigated the expression of two classes of phase II enzymes, sulfotransferases (SULT) and Uridine 5′-diphospho-glucuronosyl transferases (UGT). All tested SULTs and UGTs were found to be expressed on similar mRNA levels between the two cell types (Figure 2B). In agreement with that, incubations with the substrate 7-OH-coumarin revealed similar or higher SULT and UGT activity levels in hiPS-HEP and in hphep cultured for 20 h (including the activity assay; Figure 2E,F). Similarly to the CYP activities, phase II enzyme activities were stable or even increasing in hiPS-HEP during a 14-day culture period (between day 4–19 post-thaw).

In addition to phase I and II enzymes, transporter proteins play an important role in xenobiotics metabolism. Therefore, we also investigated the mRNA expression of 19 transporters, including both uptake and efflux transporters. 11 transporters were expressed on comparable levels in both cell types, e.g., *ABCC2* (MRP2) and *SLC10A1* (NTCP), whereas eight were expressed at lower levels in hiPS-HEP, e.g., *ABCB11* (BSEP) and *SLCO1B1* (OATP1B1; Figure 2C). The protein expression of MRP2, NTCP, BSEP, and OATP1B1 were confirmed by immunostainings (Figure 2G1–G4), however, formation of bile canaliculi was not observed when performing a 5(6)-Carboxy-2′,7′-dichlorofluorescein diacetate (CDFDA) staining (data not shown) indicating that no cell polarity is formed in hiPS-HEP cultures.

Furthermore, in order to assess batch-to-batch variation coefficient of variation (CV) was calculated for Figure 1A (CV < 8%) and for Figure 2A–C (CV < 15% for the majority (87%) of these genes). This indicates little variation between batches derived from the same hiPSC line at different occasions and thus a stable differentiation procedure.

### 2.3. Glucose Metabolism and Insulin Signaling

Another important and rapidly growing area of use for hepatocytes is disease modeling. For example, hepatocytes are crucial for modeling metabolic functions performed by the liver and for diseases such as non-alcoholic fatty liver disease (NAFLD) and non-alcoholic steatohepatitis (NASH) which are global, rapidly growing problems. In order to evaluate the utility of hiPS-HEP for modeling NAFLD/NASH, we started by investigating the energy metabolism of glucose and lipids as well as insulin signaling and inflammation in the hiPS-HEP.

First, we compared the mRNA expression of more than 40 genes central in glucose metabolism in hiPS-HEP and hphep and found the vast majority to be expressed at similar levels in both cell types (Figure 3A). Only eight of the genes were significantly differently expressed between hiPS-HEP and hphep d0 and/or d1. The genes expressed at lower levels in hiPS-HEP were fructose-1,6-biphosphatase 1 (*FBP1*), glutamic-oxaloacetic transaminase 1 (*GOT1*), pyruvate carboxylase (*PC*), and phosphoenolpyruvate carboxykinase 1 (*PCK1*), compared to hphep d0. On the other hand, Glycerol-3-phosphate dehydrogenase (*GPD2*), glycogen synthase 1 (*GYS1*), Oxoglutarate Dehydrogenase (*OGDH*), and hexokinase 2 (*HK2*) showed higher mRNA expression levels in hiPS-HEP compared to hphep d0 and/or d1. Interestingly, *PC, PCK1, FBP1, GOT1*, and *GPD2* are directly involved in gluconeogenesis, and *OGDH* has a strong influence on 2-oxoglutarate, which regulates the gluconeogenesis in the liver [18]. Noteworthy is that *PC, PCK1,* and *FBP1* were also significantly lower expressed in hphep d1 than in hphep d0.

One important hepatocyte function in glucose metabolism is the ability to synthesize and store glycogen (glycogenesis). Therefore, Periodic Acid Schiff (PAS) staining was performed on both hiPS-HEP at day 12 after thaw and on hphep at day 1 after thaw. As shown in Figure 3C1–C3, glycogen was detected in a subset of hiPS-HEP (derived from all three hiPSC lines). Similarly, hphep also showed glycogen storage in a subpopulation of the cells (Figure 3C4). The presence of glycogen is in agreement with the finding that several key enzymes of glycogenesis are well expressed in hiPS-HEP: Hexokinase (*HK*), UDP-glucose pyrophosphorylase (*UGP2*), glycogen synthase (*GYS*), and Glycogen branching enzyme (*GBE1*; Figure 3A).

Another essential feature of metabolically functional hepatocytes is the physiological response to insulin. Thus, the mRNA expression of 19 genes involved in insulin signaling were compared in hiPS-HEP and hphep and no significant differences in expression levels of these 19 genes were found between the two cell types (Figure 3B), including key genes such as insulin receptor (*INSR*), insulin receptor substrate 1 and 2 (*IRS1, IRS2*), and Ak strain transforming (AKT) serine/threonine kinases 1 and 2 (*AKT1, AKT2*). In order to test if the hiPS-HEP respond to an insulin stimulus by phosphorylation of AKT kinase which is a key regulator in the insulin signaling cascade, the hiPS-HEP were incubated for 3 h in an insulin-free medium followed by an incubation for 10 min with 100 nM insulin. In the insulin-treated cells, a significant increase of phosphorylated AKT could be detected compared to untreated control cells, while the total AKT content was similar in treated and untreated cells, as shown by Western Blot (Figure 3D).

### 2.4. Cholesterol and Lipid Metabolism

Next, we assessed the mRNA expression levels of 29 genes involved in cholesterol metabolism and 47 genes involved in fatty acid metabolism in both hiPS-HEP and hphep. The vast majority of the cholesterol metabolism-related genes assessed were expressed at similar levels in hiPS-HEP and hphep, e.g., low-density lipoprotein (LDL) receptor (*LDLR*) and Proprotein convertase subtilisin/kexin type 9 (*PCSK9*) (Figure 4A). Only four of the investigated genes differed significantly between hiPS-HEP and hphep d0/1: ATP citrate lyase (*ACLY*) and apolipoprotein A1 and A4 (*APOA1; APOA4*), which were higher in hiPS-HEP, and apolipoprotein A5 (*APOA5*) which was lower expressed in hiPS-HEP (Figure 4A). Noteworthy, the important gene *APOB* is expressed on similar level in hiPS-HEP and hphep d0. Next, we tested if the hiPS-HEP were capable of taking up fluorescently labeled low density lipoprotein (LDL-DyLight). Already after a 3-h incubation, accumulation of LDL-DyLight could be observed in the cells (Figure 4C) and an even stronger accumulation could be seen after 24 h.

The assessment of 48 genes involved in fatty acid metabolism revealed that only three genes were expressed at different levels in hiPS-HEP and hphep d0 (Figure 4B). The exceptions were acetyl-CoA carboxylase beta (*ACACB*) and bile acyl-CoA synthetase (*SLC27A5*), which were both expressed at lower levels in hiPS-HEP, and protein kinase AMP-activated catalytic subunit alpha 2 (*PRKAA2*), which was expressed at higher levels in hiPS-HEP than in hphep d0. Additionally, three genes were significantly different in hiPS-HEP compared to hphep d1; *HACD3* and *PECR* (both higher in hiPS-HEP); and *PPARD* (lower in hiPS-HEP). Noteworthy, *PECR* was also higher in hphep d1 compared to hphep d0, and *PPARD* and SLC27A5 were also lower in hphep d1 compared to hphep d0.

### 2.5. Inflammatory Response and Induction of Steatosis

Many genes involved in inflammatory response are expressed by hepatocytes and are of interest for disease modeling. Figure 5A shows the expression levels of 72 inflammation-related genes in hiPS-HEP and hphep d0/1. Noteworthy, most inflammation-related genes are expressed at low or medium levels in both hiPS-HEP and hphep d0/1, e.g., caspase 4 (*CASP4*), tumor necrosis factor (*TNF*), Toll like receptor 1, 3, 4 (*TLR1, TLR3, TLR4*) which may indicate a healthy, non-inflammatory status of both cell types under standard conditions (Figure 5A). However, a few genes are expressed at high levels in both cell types, e.g., *AIMP1, ATRN, F11R, ITCH, MIF, PLAA, PRDX5*, and *RELA*.

When comparing expression levels between hiPS-HEP and hphep d0/1, we found the majority of the genes to be similarly expressed (Figure 5A) which is similar to previous results (Figure 3 and Figure 4). Only 12 out of the 72 genes differed significantly in expression levels between the two cell types: *CCL16, CRP, FOS, LTBR*, and *ORM1* were lower expressed in hiPS-HEP, whereas *AXL, CYBA, F2R, F2RL1*, and *RIPK2* were expressed at higher levels in hiPS-HEP.

Inflammation in the liver can, amongst others, be caused by fat accumulation. At first, fat is accumulated in the hepatocytes, a condition known as NAFLD, which can progress to NASH which is characterized by steatosis, inflammation, and fibrosis [19]. To model these conditions in vitro, one prerequisite is that hepatocytes can take up free fatty acids and store as lipid droplets. Therefore, we incubated hiPS-HEP for 24 h with either 200 or 600 µM oleic acid (OA; coupled to fatty acid free bovine serum albumin (BSA)) or with fatty acid free BSA alone as vehicle control. Subsequently, the cells were stained for lipid accumulation using Oil Red O. Only a few lipid droplets were observed in the control cells (Figure 5B1) while the cells incubated with 200 and 600 µM OA showed a dose dependent increase in lipid droplets (Figure 5B3,B4, respectively). When we combined 200 µM OA with 1 µM Thapsigargin, a compound known to cause endoplasmic reticulum (ER) stress and to worsen steatosis [12], a clear increase of lipid accumulation was observed compared to only 200 µM OA (Figure 5B2,B3, respectively).

Next, we investigated if inducing steatosis by treatment with OA and Thapsigargin caused an inflammatory response in the hiPS-HEP. Indeed, we found that the mRNA expression of the inflammatory marker tumor necrosis factor α (*TNFα*) was upregulated in OA/Thapsigargin-treated hiPS-HEP (Figure 5C). The strongest upregulation of *TNFα* was caused by 600 µM OA, followed by the combination of 200 µM OA and Thapsigargin compared to vehicle control (Figure 5C). Thus, this indicated that treatment with OA and Thapsigargin caused an inflammatory response.

### 2.6. Co-Cultures of hiPS-HEP and Hepatic Stellate Cells for Modeling NASH

In the advanced stage of NASH, fibrosis occurs in the liver, which is characterized by collagen deposited by activated hepatic stellate cells (HSC). Thus, for modeling NASH-related liver fibrosis, one needs to co-culture hepatocytes and HSC. Therefore, we adapted the culture conditions in a way that allowed co-culturing primary human HSC together with hiPS-HEP. In order to activate HSC, we treated the co-cultures with 20 ng/mL transforming growth factor β (TGFβ) for 4 days which strongly increased the protein expression of α-smooth muscle actin (α-SMA) (Figure 5D2), a characteristic marker for activated HSC, compared to untreated cultures (Figure 5D1). Since activated HSC are known to express collagens, we investigated collagen mRNA expression in the co-cultures after TGFβ treatment. In accordance with the observed increase in α-SMA expression upon TGFβ stimulation (Figure 5D2), we also observed increased *Collagen 1a1* and *5a1* mRNA expression in the TGFβ-treated co-cultures compared to hiPS-HEP cultures without HSC (Figure 5D3,D4). Monocultures of hiPS-HEP alone showed no *Collagen 1a1* expression irrespective of treatment (Figure 5D3), whereas *Collagen 5a1* was expressed in untreated hiPS-HEP monocultures and upregulated upon TGFβ treatment, however at lower levels than in the co-cultures (Figure 5D4). Since TGFβ has been reported to suppress *HNF4α* expression [20] and *HNF4α* expression is crucial for maintaining a differentiated hepatocyte phenotype [21], we investigated *HNF4α* expression both on mRNA and protein level in the co-cultures. In general, *HNF4α* mRNA expression was lower in co-cultures than in hiPS-HEP monocultures independent from TGFβ-treatment which was expected and can be explained with a decreased hiPS-HEP content in the co-cultures due to the addition of HSC (Figure 5D5). Importantly, TGFβ-treatment of the co-cultures tended to decrease *HNF4α* mRNA expression levels (Figure 5D5) but the observed differences were not significant. In accordance with this, fewer HNF4α-immuno-positive nuclei could be observed the TGFβ-treated cultures (compare Figure 5D1,D2).

Taken together, the co-cultures responded as expected to TGFβ treatment with activation of the HSC and upregulation of α-SMA and collagen expression. However, already the untreated control co-cultures displayed a weak α-SMA immunostaining (Figure 5D1) which indicated that a subpopulation of the HSC was already activated. This is in agreement with previous reports that HSC get activated in 2D cultures due to the stiff matrix [22]. In order to avoid the HSC activation due to the 2D culture setting, we generated spheroids of hiPS-HEP and HSC. To this end, we seeded hiPS-HEP and HSC in ultra-low attachment plates, spheroids formed within 1–2 days and could be maintained for at least 35 days (Figure 5E1–E3). In future studies, the utility of the co-culture spheroids for NAFLD and NASH studies can be explored.

### 2.7. Similarities and Differences between hiPS-HEP and Hphep

To further assess the similarities and differences between hiPS-HEP and hphep on gene expression level, a global analysis of the transcriptomics data was carried out. Figure 6A depicts the number of transcripts that are similarly expressed, with a coefficient of variation (CV) < 0.1, throughout all samples between the different comparisons. A great majority of the transcripts (24,685 transcripts, 86%) are similarly expressed across all group comparisons. Not surprisingly, the highest number of similarly expressed genes (27,157) in the different sample group comparisons is found in hphep d0 vs. hphep d1. hiPS-HEP show a comparable number of similarly expressed genes when compared to hphep d1 and hphep d0 (26,492 and 26,244, respectively).

Using a generalized linear model (GLM) and the criterion described in the method section, in total 2866 differentially expressed genes (DEGs) were detected as significant in at least one of the sample group comparisons. Figure 6B displays the overlap of DEGs between the three group comparisons. Out of these DEGs, only 76 genes are differentially expressed between all sample group comparisons. A total of 2655 genes are differentially expressed when comparing hiPS-HEP to hphep, either at day 0 or day 1 (Figure 6B, areas a2–a7). Approximately 25% of these genes are also differentially expressed between hphep at day 1 and hphep at day 0 (Figure 6B, areas a2,a4,a6). To assess the biological relevance of the DEGs, a pathway over-representation analysis was carried out on the different groups of DEGs according to the overlapping areas (areas a1–a7) in Figure 6B. For a complete list of significant over-represented pathways, see Appendix A. Among the 76 DEGs that differ between all three groups (Figure 6B, area a4) there is an over-representation for pathways such as, “Cytochrome P450—arranged by substrate type”, “CYP2E1 reactions”, “Phase I—Functionalization of compounds”, “Phase II—Conjugation of compounds”, as well as pathways related to bile acids and bile salts. These pathways are also over-represented in the comparison between hphep d1 and hphep d0 (Figure 6B, area a1), as well as in the overlap between hphep d1 vs. hphep d0 and hiPS-HEP vs. hphep d0 (Figure 6B, areas a2,a4). The genes *CYP2E1, CYP4A11*, and *CYP2C8*, associated with the CYP P450 pathways, have the following expression rank between the groups: hphep d0 > hphep d1 > hiPS-HEP. The same expression pattern is seen for the bile acid/bile salts related genes (*SLC27A5, SLCO1B1, AKR1C4*). The genes associated with “Phase II—Conjugation of compounds” have a more ambiguous expression, *ACSM5*: hphep d0 > hphep d1 > hiPS-HEP, *SULT2A1*: hphep d0 > hiPS-HEP > hphep d1, and *UGT2A3*: hiPS-HEP > hphep d0 > hphep d1.

The DEGs in hiPS-HEP compared to hphep d1 (Figure 6B, area a7) show an over-representation for “Attenuation phase” and “HSP1-dependent transactivation”; the associated genes are *HSPA1A, HSPA1B, DNAJB1*, with the ranking hphep d1 > hphep d0 > hiPS-HEP, and *CAMK2D* with an opposite expression ranking. When comparing hiPS-HEP to hphep d0 (Figure 6B, area a3), hiPS-HEP show a higher expression of genes related to glycogen synthesis and metabolism. For the comparison hphep d0 vs. hiPS-HEP and hphep d1 vs. hiPS-HEP (Figure 6B, area a5), the pathways “Laminin interactions”, “Non-integrin membrane-ECM interactions”, “ECM proteoglycans”, and “Extracellular matrix organization” are over-represented with the expression ranking hiPS-HEP > hphep d1 > hphep d0. The DEGs in hphep d1 when compared to hiPS-HEP or hphep d0 (Figure 6B, area a6), are overrepresented in pathways related to respiratory electron transport and the citric acid cycle.

## 3. Discussion

The present study covers an in-depth characterization of hiPSC-derived hepatocytes including multiple phenotypic and functional aspects of relevance for several application areas, for example modeling of metabolic disease.

An essential requirement for the utility of stem cell-derived hepatocytes for all application areas is that cultures with a very high hepatocyte content can be robustly derived from large panels of stem cell lines. To evaluate the hepatocyte content in the hiPS-HEP cultures, we stained for the key hepatocyte transcription factor HNF4α and found that it was expressed in 92% of cells in the hiPS-HEP cultures, thus indicating a highly efficient differentiation result and a near-homogeneous culture regarding HNF4α expression. A similarly high differentiation efficiency, 93.8% HNF4α-positive cells, was previously shown when we screened a panel of 25 human pluripotent stem cell lines using an earlier version of the hepatocyte differentiation protocol used in this study to derive hiPS-HEP [16]. Notably, the hepatocyte differentiation protocol worked for all lines without any modifications. In contrast to our findings, others have reported that their protocols either do not work for some hPSC lines making it necessary to select lines that appear more prone to differentiate into hepatocytes [23], or that they require time- and labor-intensive adaptations of the protocol for individual lines [24]. One great advantage of the hiPSC technology is the possibility to generate iPSC lines from many different donors, healthy and diseased, so that one can recreate the population diversity in vitro. However, this requires differentiation protocols that, without adaptations, generate highly pure hepatocyte cultures from virtually all lines.

Furthermore, the reproducibility of the differentiation protocol is of great importance so that different cell batches show the same characteristics. We found that different hiPS-HEP batches derived from the same hiPSC line showed overall low variation measured as coefficient of variation (CV). For example, expression of hepatocyte markers (Figure 1A) show CV < 5% for all genes except for *ASGR1* (CV < 8%), and expression of the drug metabolizing machinery show CV < 15% for the majority (87%) of all genes in Figure 2A–C.

In contrast to the nearly uniform HNF4α-expression in hiPS-HEP, other markers or features are expressed only in a subpopulation of both hiPS-HEP and hphep, e.g., Albumin, α1-Antitrypsin, ASGPR1, and glycogen storage (Figure 1 and Figure 3). One would expect to find such different populations of hepatocytes since this is well-described for the liver and known as metabolic zonation [10]. In the liver lobe, hepatocytes exist in periportal and perivenous phenotypes and therefore, it is important to look at features for both populations, e.g., albumin expression (predominantly in periportal hepatocytes) and CYP expression (predominantly in perivenous hepatocytes), when assessing a hepatocyte cell model. However, we cannot rule out that the observed heterogeneity for maturation markers such as Albumin and ASGPR1 in the hiPS-HEP cultures could be due to different levels of maturity.

We assessed the maturation status of the hiPS-HEP by performing transcriptomics analysis and functional assays in comparison to adult human primary hepatocytes. Albumin secretion, which is one hallmark for adult hepatocytes, was measured at comparable levels in hiPS-HEP derived from line ChiPSC12 and hphep (CV < 9%; Figure 1D) and with slightly higher secretion in hiPS-HEP derived from line ChiPSC18 and ChiPSC22. This was confirmed by comparable *Albumin* mRNA levels across both cell types (CV < 9%; Figure 1A). Another characteristic of adult hepatocytes is high expression of the adult enzymes *CYP3A4* and low expression of the fetal enzyme *CYP3A7* which was the case in both cell types (Figure 2A). ASGPR1, a marker for stem cell-derived hepatocytes enriched for primary hepatocyte features [25], showed also comparable mRNA levels and immunostainings in hiPS-HEP and hphep (Figure 1A,C3,C7) which suggested that hiPS-HEP have several adult hepatocyte features at levels comparable to hphep.

Two functionalities that require further improvement in hiPS-HEP are, (1) urea secretion and (2), the drug metabolizing machinery, comprising phase I and II enzymes as well as transporters. Urea secretion was clearly lower in hiPS-HEP than in hphep d1 (Figure 1E). The only urea cycle gene that was lower in hiPS-HEP than in hphep was *ARG1* (Figure 1B) which is the last in the chain of five key enzymes in the urea cycle [26] and is likely the cause of the restricted urea production. Regarding phase I enzymes, the three main CYP enzymes, CYP1A, 3A, and 2C9, had comparable activity levels in hiPS-HEP and in hphep d1. CYP1A, 3A, and 2C9 combined metabolize about 60% of prescription drugs (CYP3A4 37%; CYP1A2 9%; CYP2C9 17%; [27]), but in order to be useful for drug metabolism studies all enzymes need to be expressed at useful levels. Therefore, expression of CYP2B6, 2D6, and 2C19 needs to be improved in hiPS-HEP (Figure 2D). We observed a similar situation for transporters as for the CYP enzymes: some were well expressed, e.g., MRP2 (*ABCC2*) and NTCP (SLC10A1) and some need to be increased in hiPS-HEP, e.g., *OATP1B1* and *OATP1B3* (Figure 2C). In addition, formation of bile canaliculi was not observed, meaning that hiPS-HEP do not develop cell polarity with apical and basolateral membrane compartments. This limits the utility of the cells for applications requiring bile canaliculi formation, e.g., biliary excretion of drugs. Regarding phase II metabolism, we focused on UGTs and SULTs since they are responsible for more than half of all phase II metabolism of clinically used drugs [28] and found SULT and UGT activity levels to be comparable in hiPS-HEP and hphep (Figure 2E,F).

Noteworthy, Albumin and urea secretion as well as CYP and phase II activities were stable over time in hiPS-HEP (Figure 1D–E; Figure 2D–F). This phenotypic stability of the hiPS-HEP is clearly a superior feature over hphep, which are known to quickly loose functions such as CYP activities in conventional 2D cultures [29,30]. Many studies, for example for hepatitis virus studies and toxicity studies with repeated dosing, require a chronic treatment and a culture period of at least 2 weeks which is why phenotypic stability is of great importance for an in vitro hepatocyte model.

As expected, hphep display large variations of CYP activities between donors (see error bars for hphep in Figure 2D) which reflect the large inter-individual variation existing in the population (see e.g., [17]). This variation is a major issue that researchers are facing when using primary hepatocytes for in vitro models since the amount of hphep from one donor is limited and researchers are therefore forced to switch to new donors that may have a very different phenotype potentially affecting their studies. Therefore, one main advantage of hiPS-HEP is that one can obtain virtually unlimited amounts of cells from one donor, and importantly, there is only very low variation between different batches of hiPS-HEP, as seen for example in the reproducible and robust expression of hepatocyte markers (Figure 1A), drug metabolizing genes (Figure 2A–C), CYP activities (Figure 2D), and the response to insulin (Figure 3D). Noteworthy, inter-individual variation in the CYP activity profile can be observed when comparing hiPS-HEP derived from different iPSC lines: e.g., hiPS-HEP derived from ChiPSC18 have higher CYP3A and 2D6 activities than hiPS-HEP derived from the other 2 hiPSC lines (Figure 2D2,D5), which is likely due to higher expression of the polymorphic genes CYP3A5 and 2D6 in ChiPSC18 (Figure 2A).

Energy metabolism is another key function of the liver. The results presented here indicate that hiPS-HEP have a comparable glucose and lipid metabolism as hphep, the gold standard (Figure 3 and Figure 4). However, several genes related to gluconeogenesis, e.g., *PC, PCK1, FBP1, GOT1, GPD2,* and *OGDH*, are expressed on lower levels in hiPS-HEP (Figure 3A). A potential cause could be the relatively high glucose concentration in the culture medium which would also explain the downregulation of gluconeogenesis-related genes in hphep d1 compared to hphep d0 (Figure 3A).

One important area of use for hepatocytes is to model NAFLD, a common cause for chronic liver disease with a prevalence of 20–30% in Europe [31]. NAFLD can progress to NASH which is associated with a higher risk of liver-related mortality and cardiovascular disease [32], and further to cirrhosis with an increased risk for hepatocellular carcinoma [33]. The long progression until clinical signs, the lack of good diagnostic tools as well as the absence of relevant preclinical models hampers the development of adequate treatments and drugs for NADFL/NASH, and highlights the need for novel translational in vitro models [34]. Several animal models have been developed, however, due to the heterogeneous pathology few or none of these models represent the human situation accurately [35]. Therefore, there is great and urgent need for an in vitro model for NAFLD/NASH.

Our data presented here suggest that hiPS-HEP could serve as such a potential NAFLD/NASH model. Most genes related to lipid and cholesterol metabolism are expressed at comparable levels as in hphep d0/1 (Figure 4). Only four genes differed significantly between hiPS-HEP and hphep d0/1, for example *APOA5*, which is involved in lipoprotein metabolism by interacting with LDL receptor [36]. However, despite the lower *APOA5* expression in hiPS-HEP and in agreement with substantial expression of the *LDL receptor*, we found that hiPS-HEP accumulated labelled LDL upon a short incubation (Figure 4A,C). We also observed dose-dependent steatosis, which in turn lead to dose-dependent upregulation of the inflammatory marker *TNFα* (Figure 5B,C). Importantly, under standard culture conditions most inflammation related genes were expressed at low levels both in hphep and hiPS-HEP (Figure 5A), indicating a healthy status of the cells and no activation of an inflammatory response which is a prerequisite for the utility of the hiPS-HEP for modelling inflammation. Furthermore, hiPS-HEP responded in the anticipated manner to Thapsigargin-induced ER stress [12,37], with both an increased lipid accumulation and an upregulated *TNFα* expression (Figure 5B,C). Taken together, these results indicate that hiPS-HEP are comparable to adult primary hepatocytes regarding glucose and lipid metabolism as well as inflammatory markers. The observed responses are essential for a relevant human in vitro model for NAFLD/NASH and highlight the potential use of hiPS-HEP for such models. Another important finding is that key drug targets for NAFLD/NASH treatment are well expressed in hiPS-HEP, e.g., the peroxisome proliferator−activated receptor-α (*PPARA*) and -δ (*PPARD*), which are targets for the drug elafibranor [38], and the nuclear receptor FXR (*NR1H4*), which is a target for the drug obeticholic acid [39]. All three genes are expressed on the same level in hiPS-HEP and hphep d0 (Figure 4B) which demonstrates the potential of hiPS-HEP for drug target studies for NAFLD/NASH.

Since the progression of NAFLD to NASH includes a contribution of inflammatory and fibrogenic response from non-parenchymal cells, hepatocytes need to be co-cultured with non-parenchymal cells, such as stellate cells and Kupffer cells, as reviewed in [35]. To date, only a limited number of attempts using primary cells or cell lines (reviewed in [40]) or hepatocytes derived from human pluripotent stem cells [12,41] have been reported in the NASH field. However, in the two NASH studies utilizing stem cell-derived hepatocytes, those were not co-cultured with non-parenchymal cells [12,41]. Two other studies described 2D co-cultures of several hiPSC-derived liver cell types: liver progenitor cells and HSC [42], and liver progenitor cells, liver sinusoidal endothelial cells and HSC [43], respectively, and reported a positive effect of the co-culture on the maturity of the hiPSC-liver progenitor cells but did not evaluate the utility of the co-cultures for modeling NASH. Another study reported modeling of liver fibrosis using the hepatoma cell line HepaRG and hiPSC-derived HSC [44]. To the best of our knowledge, our study is the first one to describe co-cultures of hiPSC-derived hepatocytes and human primary HSC. Importantly, we found that α-SMA expression, a hallmark of activated HSC and fibrosis, was induced in HSC in 2D co-cultures by TGFβ treatment. Since HSC showed a weak activation in 2D even without TGFβ stimulation, we generated 3D-spheroids of hiPS-HEP and HSC which can be maintained for at least 35 days. The results presented here pave the way for further exploring the utility of these co-culture spheroids for NASH studies. The 3D co-culture setting may even further improve the hiPS-HEP functionality compared to 2D cultures, similarly to reports on 3D spheroids consisting of hphep and non-parenchymal cells [8].

The global analysis of the transcriptomics data revealed very similar phenotypes of the three cell populations (Figure 6A). Differences were observed, amongst others, in phase I and II metabolism, indicating a lower functionality of hiPS-HEP and hphep d1 compared to hphep d0, which is in line with our gene expression and functional data on drug metabolism (Figure 2). Interestingly, in the DEGs between hiPS-HEP and hphep d0 and d1 “Attenuation phase” and “HSP1-dependent transactivation” are over-represented pathways which are related to cellular response to heat stress. It has been reported previously that cryopreservation and re-plating of hphep results in reduced ability to form cell-matrix and cell-cell interactions which triggers stress and heat shock responses [45]. Our observation that DEGs involved in these pathways (*HSPA1A, HSPA1B*, and *DNAJB1*) are upregulated in hphep d0 and d1 compared to hiPS-HEP is in line with these reports. Furthermore, pathways connected to ECM and laminin interactions are over-represented in the DEGs between hiPS-HEP and hphep d0 and d1, with higher expression in hiPS-HEP. A possible explanation could be that hiPS-HEP are differentiated in 2D cultures whereas hphep come from a 3D environment. It would be interesting to investigate if these pathways are downregulated in hiPS-HEP 3D spheroids.

The identification of DEGs is of great importance since they can provide a basis for future improvements as the aim is to reach the same functionality and maturity in hiPS-HEP as in hphep d0. One incentive for such continuous efforts is that hphep are not useful as a cell source for gene editing and are limited for disease modeling of a specific genetic background. The advance of the hiPSC technology allows to routinely generate hiPSC lines from many individuals, including healthy persons and patients, and thus represent the whole population. Furthermore, genetic modification of hiPSC lines using for example CRISPR/Cas9, allows to study the molecular basis of diseases.

Taken together, the hiPS-HEP used in this study represent a homogenous cell population with a high similarity to hphep and a potential for modeling NAFLD/NASH. Our results highlight the potential of hiPSC-derived hepatocytes as a virtually unlimited cell source. In order to build more advanced cell models, to better recapitulate the human liver environment, to reflect the population diversity and to increase the predictability for disease modeling, advanced 3D culture systems including multiple cell types and microfluidics as well as panels of hiPSC lines derived from various ethnicities need to be developed and implemented.

## 4. Materials and Methods

### 4.1. Cell Culture

Cryopreserved hiPS-HEPs derived from the hiPSC lines ChiPSC12, ChiPSC18, and ChiPSC22 were thawed, plated, and maintained according to the vendor instructions (Cellartis Enhanced hiPS-HEP v2 kits, Cat.no. Y10133, Y10134, Y10135, Takara Bio Europe, Gothenburg, Sweden, www.takarabio.com). Enhanced hiPS-HEP v2 were maintained for up to 21 days post-thawing with media changes every 2 or 3 days using the Cellartis Enhanced hiPS-HEP Long-Term Maintenance Medium included in the kits.

For 2D co-cultures of Enhanced hiPS-HEP v2 and primary human stellate cells (HCS, BioIVT, Brussels, Belgium), cells were seeded on a Fibronectin coating (concentration 2 µg/cm^2^; Roche, Basel, Swiss) in a 2:1 ratio (hiPS-HEP: HSC) at a cell density of 400 K/cm^2^ using the regular plating medium included in the hiPS-HEP v2 kits. Instead of the regular HEP Long-Term Maintenance Medium the following maintenance medium was used for the co-cultures: Williams’ Medium E supplemented with 0.1% PEST (15140–130, ThermoFisher, Stockholm, Sweden), 0.5% DMSO (D2650, Sigma, Darmstadt, Germany), 0.67% Cellartis Hepatocyte supplement (Y11149, Takara Bio), 2% Cellartis Hepatocyte Additives (Y11078, Takara Bio), 194 µM L-Proline (P5607, Sigma), and 173 µM L-ascorbic acid 2-phosphate sesquimagnesium salt (A8960, Sigma).

For 3D co-cultures of Enhanced hiPS-HEP v2 and HSC, cells were seeded in a 2:1 ratio (hiPS-HEP: HSC) in InVitroGRO CP medium (BioIVT) supplemented with 0.5% PEST, and 2% Cellartis Hepatocyte Additives (Y11078, Takara Bio) in Costar 96 well ultra-low attachment plates (7007, Corning, Amsterdam, The Netherlands) with 30,000 cells per well in 100 µL plating medium. At 24 h post-seeding, medium was changed to 3D spheroid maintenance medium as described by Rashidi and colleagues [46]. At 24–48 h post-seeding, spheroids had formed and were maintained for up to 35 days with 50% medium changes every 2–3 days. Alternatively, spheroids could also be maintained with the 2D co-culture medium described above.

Cryopreserved hphep from three donors (Cat.no. M00995-P for male donors, F00995-P for female donors, BioIVT), selected based on high viability (see Table 1 for donor details), were thawed and plated according to the provider’s instructions. For experiments that required the hepatocytes to be cultured for 24 h, medium was changed 4 h post-plating to fresh plating medium (InVitroGRO CP medium, BioIVT).

### 4.2. Transcriptomics Analysis

#### 4.2.1. Total RNA Extraction

Two batches of Enhanced hiPS-HEP v2 from each hiPSC line (ChiPSC12, ChiPSC18, and ChiPSC22) were harvested on day 13 post-thawing by scraping the cells in the culture medium, centrifuging at 300× *g* for 5 min and frozen as dry cell pellets. Human primary hepatocytes (*n* = 3 donors) were harvested as dry cell pellets directly after thawing (day 0) and on day 1 post-thawing (day 1). Cell pellets were stored at −80 °C until RNA extraction. Total RNA was extracted from the cell pellets using the GenElute RNA/DNA/protein Plus Purification kit (E5163, Sigma Aldrich). RNA amounts were quantified using a NanoDrop ND-1000 (NanoDrop, http://www.nanodrop.com).

#### 4.2.2. Gene Array

The quality of the RNA and cDNA was verified using a 2100 Agilent Bioanalyzer. To measure the mRNA levels, cDNA was synthesized from the RNA samples applying the GeneChip WT PLUS Reagent Kit (Affymetrix, Stockholm, Sweden) and fragmented cDNA was hybridized at 45° Celsius for 16 h to whole transcript HuGene ST 2.0 arrays (Affymetrix, http://www.affymetrix.com) at SCIBLU Genomics (Lund University, Sweden). In total, 12 expression microarrays were run.

#### 4.2.3. Data Processing

Raw microarray data was imported into R (version 3.6.1, R Core Team, 2019, https://www.r-project.org/) and signal intensities normalized by the Robust Multichip Average method in the oligo package [47]. The two biological replicates for each hiPSC line were summarized by mean expression. To remove transcripts with expression values close to background, probes with a log_2_ expression below 5 in all samples were removed. The resulting dataset contained 21,427 transcripts and 9 samples (3 hphep day 0, 3 hphep day 1, 3 hiPS-HEP). The transcripts were mapped to Human Genome Organization Gene Nomenclature Committee (HGNC) symbols using the HuGene 2.0 ST V1 NetAffx file from Affymetrix (NA36, genome build hg19, http://www.affymetrix.com/analysis/index.affx). The microarray data used in this study follow the MIAME standard and raw expression data are available at ArrayExpress (https://www.ebi.ac.uk/arrayexpress/), accession number E-MTAB-8286.

#### 4.2.4. Statistical Analysis

Inspection of transcriptomics data revealed an approximate gamma distribution, and thus statistical testing for differential expression was based on a generalized linear model (GLM) from the gamma family with a linear link. The GLM was fitted using genes as response variables and sample group (hphep day 0, hphep day 1, hiPS-HEP) as covariates. The primary human hepatocytes day 0 and day 1 samples were treated as paired. Statistical significance of differential expression was assessed with the likelihood ratio test. *p*-values were adjusted for multiple testing by Benjamini-Hochberg correction. Differentially expressed genes were identified using a combined criterion of adjusted *p*-value < 0.05 and absolute log_2_ fold change > 2.

Pathway over-representation analysis of the differentially expressed genes was carried out with Reactome Pathway Database [48,49], using a criterion of a *p*-value < 0.05, and at least two differentially expressed genes present for the identified over-represented pathways.

### 4.3. Immunocytochemistry

Cells were stained as described previously in [7]. Briefly, Enhanced hiPS-HEP v2 were fixed on day 12 post-thawing and hphep on day 1 post-thawing, by 15 min incubation with 4% Formaldehyde. Cells were stained with the following primary and secondary antibodies: rabbit anti-α1AT (A0012, DAKO, 1:200 dilution), rabbit anti-albumin (A0001, DAKO, 1:1000), mouse anti-ASGPR1 (MAI-40244, ThermoFisher 1:50), rabbit-anti-BSEP (purchased from Bruno Stieger, University Hospital Zurich, Zurich, Switzerland, 1:100); rabbit anti-HNF4α (SC-8987, SantaCruz Biotechnology, 1:300, Heidelberg, Germany), rabbit anti-MRP2 (SC-20766, SantaCruz Biotechnology, 1:50), rabbit anti-NTCP (Bruno Stieger, 1:400), rabbit anti-OATP1B1 (Bruno Stieger, 1:200), donkey anti-rabbit Alexa 594 IgG (A21207, ThermoFisher, 1:1000) or goat anti-mouse Alexa 488 (A11029, ThermoFisher, 1:1000). For nuclear counterstaining, DAPI (Sigma, D9542) was added during the incubation with the secondary antibodies (add 2 µL/mL of a 1 mg/mL DAPI stock in DMSO). Stainings were examined using an inverted fluorescence microscope (Eclipse Ti-U, Nikon, Amsterdam, The Netherlands), ANDOR Zyla sCMOS digital camera and the NIS-Elements software package (version 4.30). Technical control staining without primary antibodies were performed for all secondary antibodies and these were negative. Quantification of HNF4α positive cells was done in relation to DAPI stained cells using the CellC Cell Counting Software [50].

Co-cultures of Enhanced hiPS-HEP v2 and HSC were permeabilized with 0.5% Triton X-100 solution in PBS for 1 h at room temperature (RT) and then incubated for 30 min at RT with blocking buffer containing 1% BSA and 10% normal goat serum (Sigma). Primary and secondary antibodies were diluted in blocking buffer. Cells were incubated for 2 h at RT (α-SMA) or overnight at 4 °C (HNF4α) with the primary antibodies and subsequently washed three times with PBS before incubating with the secondary antibodies for 1 h at RT. Next, cells were washed three times with PBS, stained for 5 min with DAPI diluted 1:25,000 in PBS and washed again. The following primary and secondary antibodies were used: mouse anti-α-smooth muscle actin (α-SMA; 1:1000; A5228, Sigma), mouse anti-HNF4α (1:200; ab41898, Abcam), goat anti-mouse Alexa Fluor 594 (1:300; ThermoFisher Scientific), and goat anti-mouse FITC (1:50; ThermoFisher Scientific). Fluorescent images were acquired on an Axio Observer Z1 Zeiss microscope (Zeiss, Breda, The Netherlands) and processed with ZEN 2.3 lite software (Zeiss).

### 4.4. Period Acid-Schiff (PAS) Staining

Glycogen storage was visualized by PAS staining of Enhanced hiPS-HEP v2 derived from C12, C18, and C22 (fixed on day 12 post-thawing) and hphep (fixed on day 1 post-thawing). Cells were stained with the glycogen assay kit (MAK016, Sigma) according to the manufacturer’s protocol.

### 4.5. Albumin Secretion

Albumin secretion from Enhanced hiPS-HEP v2 (derived from C12, C18, and C22) was analyzed on days 4, 6, 12, and 20 post-thawing and from hphep cells 24 h post-thawing. The culture medium was collected after 24 h of conditioning and Albumin content was analyzed with the Albuwell kit (Exocell, Philadelphia, PA) according to the manufacturer’s protocol. The Albumin content in the medium was normalized to the assay duration (24 h) and the amount of protein per well.

For protein quantification, cells were washed once with DPBS (with Calcium and Magnesium) and lysed in 0.02 mM NaOH over night at 4 °C and stored at −20 °C until quantification using the Pierce BCA Protein Assay kit (ThermoFisher, Rockford, IL) according to the manufacture’s instruction.

### 4.6. Urea Secretion

On days 13 and 20 post-thawing, Enhanced hiPS-HEP v2 and hphep d1 were incubated with 5 mM ammonium chloride for 24 h. After 24 h, medium was collected, and urea secretion was analyzed with the QuantiChrom Urea Assay Kit (BioAssay Systems, Hayward, CA, USA). Urea content was normalized to the amount of protein per well (determined using the Pierce BCA Protein Assay Kit, see above) and the assay duration (24 h).

### 4.7. AKT Western Blot

Phosphorylated AKT: Enhanced hiPS-HEP (derived from C18, on day 12 post-thawing) were incubated in insulin-free medium (phenol-red free Williams’ medium E containing 0.1% PEST, 25 mM HEPES, 2 mM L-Glutamine) for 3 h and then treated for 10 min with 0 nM or 100 nM insulin. Phosphorylated AKT (Cell Signaling) and total AKT (Cell Signaling) were quantified by Western blot (NuPAGE 4–12% Bis-Tris Protein Gels, Thermo Fisher Scientific).

### 4.8. CYP Activity Assay

The CYP activities of Enhanced hiPS-HEP v2 were analyzed by performing a CYP activity assay at days 4, 12, and 19 after thawing and the results were compared to hphep cultured for 20 h. Briefly, the cells were carefully washed twice with pre-warmed Williams’ medium E (Phenol-red free, +0.1% PEST). Then, the activity assay was started by adding 110 μL/cm^2^ culture area of pre-warmed Williams’ medium E (phenol-red free) containing 0.1% PEST, 25 mM HEPES (H7523, Sigma), 2 mM L-Glutamine, and the probe substrate cocktail (see Table 2 below). After a 2-h incubation at 37 °C, 100 μL of the supernatant was collected and kept at −80 °C until LC/MS analysis. LC/MS analysis (performed at Pharmacelsus GmbH, Saarbrücken, Germany) was used to measure the formation of the specific metabolites Acetaminophen (CYP1A), OH-Bupropion (CYP2B6), 4-OH-Diclofenac (CYP2C9), 4-OH-Mephenytoin (CYP2C19), OH-Bufuralol (CYP2D6), and 1-OH-Midazolam (CYP3A). The metabolite concentrations were normalized to the amount of protein per well (determined using the Pierce BCA Protein Assay Kit, see above) and the assay duration (120 min). To be able to normalize the results between different LC/MS runs, a metabolite cocktail with known concentrations of all metabolites is included in every analysis batch.

### 4.9. Phase II Enzyme Activity Assay

The phase II enzyme activities of Enhanced hiPS-HEP v2 were analyzed day 4, 8, 12, and 19 after thawing by performing a phase II activity assay. Briefly, the cells were carefully washed twice with pre-warmed Williams’ medium E (+0.1% PEST). Then the activity assay was started by adding 110 μL/cm^2^ culture area of pre-warmed Williams’ medium E containing 0.1% PEST, 25 mM HEPES, 2 mM L-Glutamine, and 200 μM 7-OH-coumarin. After a 2 h incubation at 37 °C, 100 μL of the supernatant was collected and kept at −80 °C until LC/MS analysis. LC/MS analysis (performed at Pharmacelsus GmbH) was used to measure the formation of 7-OH-coumarin sulfate and 7-OH-coumarin glucuronide, specific metabolites for sulfotransferases and UDP-glucuronosyltransferases, respectively. The metabolite concentrations were normalized to the amount of protein per well (determined using the Pierce BCA Protein Assay Kit, see above) and the assay duration (120 min).

### 4.10. LDL Uptake

In order to determine the uptake of low-density lipoproteins (LDL), Enhanced hiPS-HEP v2 (derived from C12, C18, and C22), on day 6 after thawing, were incubated for 3 h with LDL-DyLight (Cat.no. 10011125, Cayman Chemical, Hamburg, Germany) diluted 1:100 in regular maintenance medium. Next, cells were washed once with DPBS (with Calcium and Magnesium) and immunofluorescence was recorded as described above under immunocytochemistry.

### 4.11. Fatty Acid Accumulation and Inflammatory Response

Enhanced hiPS-HEP v2 were incubated for 24 h (on days 5–6 post-thawing) with Williams’ medium E containing 0.1% PEST, 25 mM HEPES, and 2 mM L-Glutamine supplemented with either 200 µM oleic acid (OA, O1008, Sigma) coupled to 77 µM fatty acid-free BSA (FAF-BSA, A8806, Sigma), 600 µM OA coupled to 231 µM FAF-BSA, or only 231 µM FAF-BSA (as vehicle control). In addition, one group of cells was treated with 200 µM OA (coupled to 77 µM FAF-BSA) plus 1 µM Thapsigargin (T0933, Sigma). After the 24 h incubation, cells were either stained with Oil Red O or harvested for gene expression analysis. Oil Red O staining was performed using the Hepatic Lipid Accumulation/Steatosis Assay Kit (Cat.no. ab133131, Abcam) according to the protocol provided with the kit. Stainings were evaluated using a Zeiss AxioVert microscope, an Axicam 105 color camera and the ZEN2 software (all from Carl Zeiss, Jena, Germany). For gene expression analysis, cells were harvested in RNAprotect Cell Reagent (Cat No. 76526, Qiagen, Hilden, Germany). RNA preparation, cDNA synthesis, and qPCR were performed as described previously in [7]. Gene expression was analyzed using the TaqMan Gene Expression Assays (Applied Biosystems, Foster City, CA): *TNFα* (Hs00174128_m1), and *CEBPα* (Hs00269972_s1) which served as a reference gene.

### 4.12. RNA Prep and RT-qPCR of Co-Cultures and 3D Spheroids

Total RNA from co-cultures and from 3D spheroids was isolated using the Ambion RNAqueous RNA Isolation Kit (ThermoFisher). The manufacturer’s supplied protocol was followed during the isolation and RNA concentrations were measured using the nanodrop ND-1000 spectrophotometer. Cellular RNA (maximum of 200 ng) was reverse transcribed into cDNA using the High Capacity RNA to cDNA kit (Applied Biosystems) in a total reaction volume of 20 µL consisting of 9 µL assay buffer, 1 µL enzyme mixture and added up to 20 µL with RNAse free H2O. Samples were incubated for 60 min at 37 °C and 5 min at 95 °C using a BioRad iCycler. qPCR samples were subsequently prepared by diluting cDNA samples 1:1 with RNAse free H2O and samples consisted of 2 µL diluted cDNA, 0.625 µL assay on demand (AoD, ThermoFisher) or 25 µM primers with 6.25 µL Taqman^®^ Gene expression mastermix (ThermoFisher) filled up to a total volume of 12.5 µL with RNAse free H2O. Samples were run at 50 °C for 2 min, 95 °C for 10 min and 40–45 cycles, each cycle consisting of 95 °C for 15 s and 60 °C for 1 min using the QuantStudio6 flex. The following TaqMan Gene Expression Assays were used: *Collagen 1a1* (Hs01076777_m1), *Collagen 5a1* (Hs00609133_m1), *HNF4α* (Hs00230853_m1), and *HPRT1* (Hs03929098_m1). The latter served as a reference gene.

## Figures and Tables

**Figure 1 ijms-21-00469-f001:**
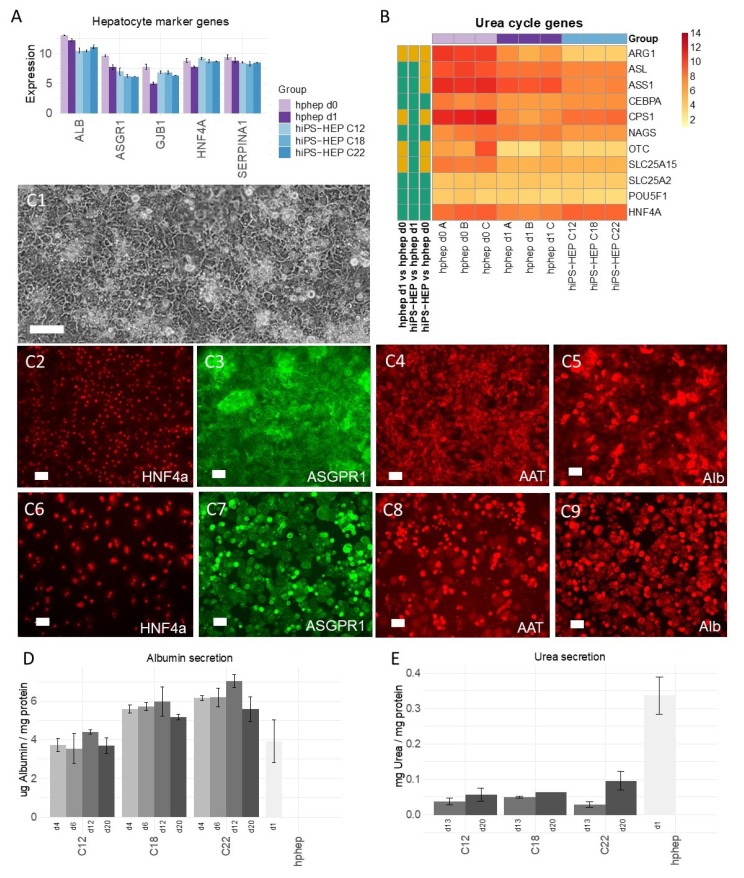
Expression of general hepatocyte markers and functions in human induced pluripotent stem cell-derived hepatocytes (hiPS-HEP). (**A**) mRNA expression of the general hepatocyte markers albumin (*ALB*), asialoglycoprotein receptor 1 (*ASGPR1*), connexin 32 (*GJB1*), hepatocyte nuclear factor 4α (*HNF4α*), and α1-antitrypsin (*AAT*) in hiPS-HEP derived from three human induced pluripotent stem cell (hiPSC) lines (ChiPSC12, ChiPSC18, ChiPSC22) on day 13 post-thawing and human primary hepatocytes (hphep) directly after thawing (d0) and on day 1 post-thawing (d1). Error bars represent standard deviation of different cell batches for hiPS-HEP (*n* = 2 batches) and different donors for hphep (*n* = 3 donors), respectively. (**B**) Heatmap of mRNA expression of urea cycle genes in hiPS-HEP derived from three hiPSC lines (ChiPSC12, ChiPSC18, ChiPSC22) on day 13 post-thawing (*n* = 2 batches per hiPSC line) and hphep directly after thawing (d0) and on day 1 post-thawing (d1). The panel to the left of the heatmap indicates significant differences between the groups (adj. *p*-value < 0.05, absolute log_2_ fold change > 2; green = no significant difference, orange = significant difference). (**C**) Representative phase contrast picture of hiPS-HEP derived from ChiPSC18 on day 4 post-thawing (**C1**), and pictures of immunocytochemistry (ICC) stainings of the hepatocyte markers Hepatocyte Nuclear Factor 4α (HNF4α), Asialoglycoprotein receptor 1 (ASGPR1), α1-Antitrypsin (AAT), and Albumin (Alb) in hiPS-HEP derived from ChiPSC18 on day 12 post-thawing (**C2**–**C5**) and hphep cultured for 24 h (**C6**–**C9**). Scale bars 100 µm in (**C1**) and 50 µm in (**C2**–**C9**). (**D**) Albumin secretion in hiPS-HEP derived from three hiPSC lines (ChiPSC12, ChiPSC18, ChiPSC22) on days 4, 6, 12, and 20 post-thaw compared to hphep cultured for one day (d1) post-thaw. Error bars represent standard deviation of duplicate wells per group. (**E**) Urea secretion in hiPS-HEP derived from three hiPSC lines (ChiPSC12, ChiPSC18, ChiPSC22) on days 13 and 20 post-thaw compared to hphep cultured for one day (d1) post-thaw. Error bars represent standard deviation of duplicate wells per group.

**Figure 2 ijms-21-00469-f002:**
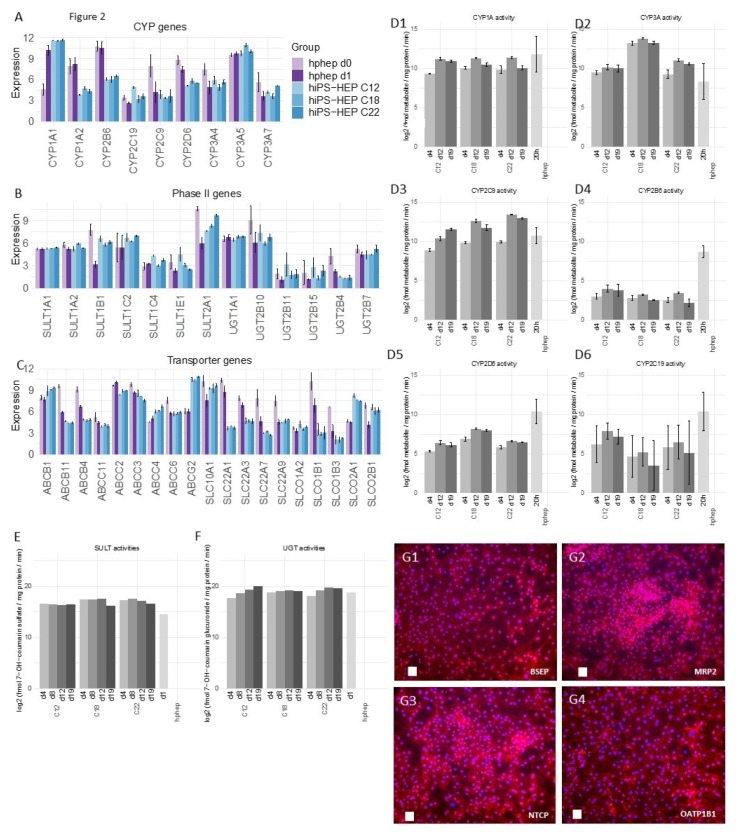
Expression of genes of the drug metabolizing machinery in hiPS-HEP. (**A**–**C**) mRNA expression of nine cytochrome P450 enzymes (**A**), 13 phase II enzymes (**B**), and 19 transporter proteins (**C**) in hiPS-HEP derived from three hiPSC lines (ChiPSC12, ChiPSC18, ChiPSC22) on day 13 post-thawing and hphep directly after thawing (d0) and on day 1 post-thawing (d1). Error bars represent standard deviation of different cell batches for hiPS-HEP (*n* = 2 batches) and different donors for hphep (*n* = 3 donors), respectively. (**D**) Enzyme activities of CYP1A ((**D1**); metabolite: Paracetamol), CYP3A ((**D2**); metabolite: OH-Midazolam), CYP2C9 ((**D3**); metabolite: OH-Diclofenac), CYP2B6 ((**D4**); metabolite: OH-Bupropion), CYP2D6 ((**D5**); metabolite: OH-Bufuralol), and CYP2C19 ((**D6**); metabolite: OH-Mephenytoin) in hiPS-HEP derived from three hiPSC lines (ChiPSC12, ChiPSC18, ChiPSC22) on days 4, 12, and 19 post-thaw compared to hphep cultured for 20 h post-thaw as measured by LC/MS. Error bars represent standard deviation of different cell batches for hiPS-HEP (*n* = 2 batches) and different donors for hphep (*n* = 3 donors), respectively. (**E**–**F**) Enzyme activities of the phase II enzymes Sulfotransferases ((**E**); metabolite: 7-OH-coumarin sulfate) and Uridine 5′-diphospho-glucuronosyl transferases ((**F**); metabolite: 7-OH-coumarin glucuronide) in hiPS-HEP derived from three hiPSC lines (ChiPSC12, ChiPSC18, ChiPSC22) on days 4, 8, 12, and 19 post-thaw compared to hphep cultured for one day (d1) post-thaw as measured by LC/MS. Data for hphep are mean values of three different donors. hiPS-HEP samples were pooled from two individual wells. (**G**) Representative pictures of ICC stainings of the transporters Bile Salt Export Pump (BSEP, ABCB11), Multidrug resistance-associated protein 2 (MRP2, ABCC2), Na+-taurocholate co-transporting polypeptide (NTCP, SLC10A1), and organic anion transporting polypeptide 1B1 (OATP1B1, SLCO1B1) in red in hiPS-HEP derived from ChiPSC18 on day 12 post-thawing (**G1**–**G4**). Nuclear counterstaining with DAPI in blue. Scale bars 50 µm. Abbreviations: CYP: Cytochrome P450; DAPI: 4′,6-diamidino-2-phenylindole; LC/MS: liquid chromatography/mass spectrometry, SULT: sulfotransferases, UGT: Uridine 5′-diphosphoUDP-glucuronosyl transferases.

**Figure 3 ijms-21-00469-f003:**
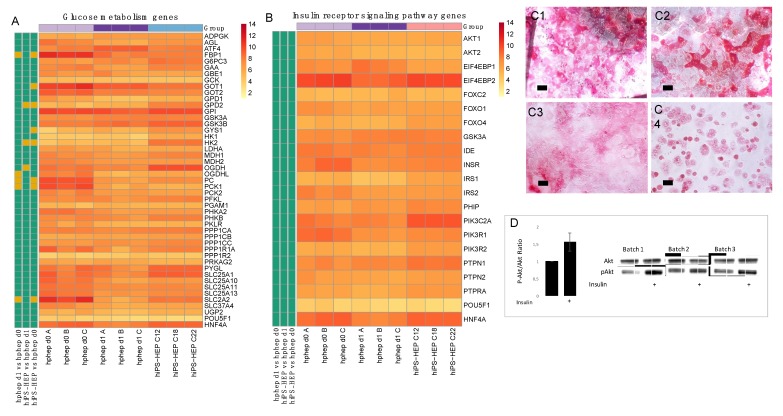
Glucose metabolism and insulin receptor signaling in hiPS-HEP. (**A**–**B**) Heatmap of mRNA expression of genes involved in glucose metabolism (**A**) and insulin signaling (**B**) in hiPS-HEP derived from three hiPSC lines (ChiPSC12, ChiPSC18, ChiPSC22) on day 13 post-thawing and hphep directly after thawing (d0) and on day 1 post-thawing (d1). The panel to the left of the heatmap indicates significant differences between the groups (adj. *p*-value < 0.05, absolute log_2_ fold change > 2; green = no significant difference, orange = significant difference). (**C**) Representative pictures of periodic acid Schiff staining of Glycogen deposition in hiPS-HEP derived from ChiPSC12 (**C1**), ChiPSC18 (**C2**), and ChiPSC22 (**C3**) on day 12 post-thawing and hphep cultured for 24 h (**C4**). Scale bars 50 µm. (**D**) Western blot detection of Ak strain transforming (AKT) and phosphorylated AKT (pAKT) in hiPS-HEP after a 10 min incubation with 100 nM insulin (as indicated by plus sign) or without 100 nM insulin, respectively. The ratio of pAKT and total AKT was calculated for three batches of hiPS-HEP derived from ChiPSC18 on day 12 post-thawing and is presented as fold change compared to the control cells without insulin stimulation. Error bars represent standard deviation of different cell batches for hiPS-HEP (*n* = 3 batches).

**Figure 4 ijms-21-00469-f004:**
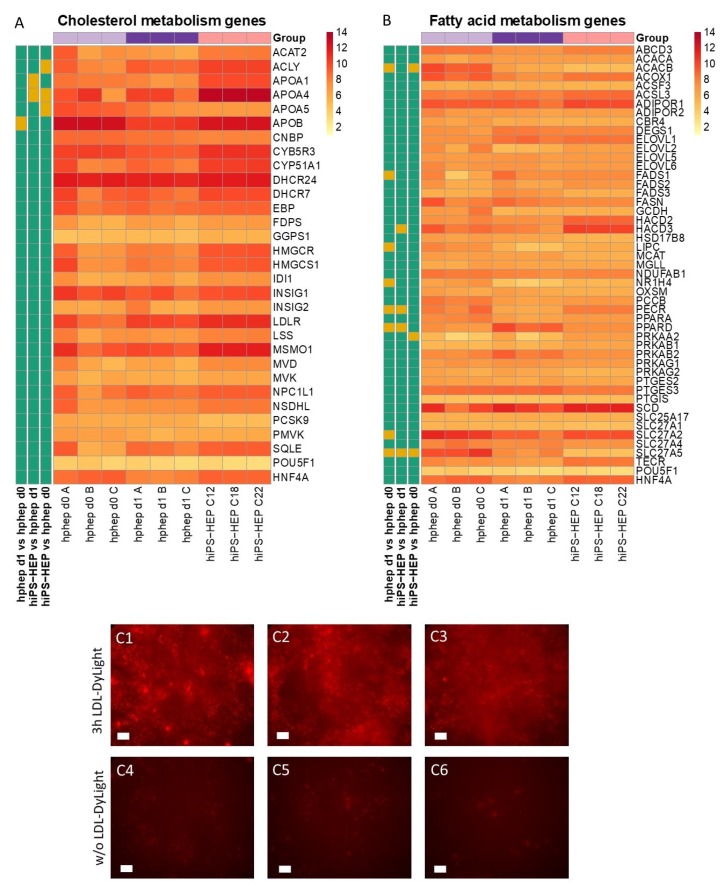
Lipid metabolism in hiPS-HEP. (**A**–**B**) Heatmap of mRNA expression of genes involved in cholesterol metabolism (**A**) and fatty acid metabolism (**B**) in hiPS-HEP derived from three hiPSC lines (ChiPSC12, ChiPSC18, ChiPSC22) on day 13 post-thawing and hphep directly after thawing (d0) and on day 1 post-thawing (d1). The panel to the left of the heatmap indicates significant differences between the groups (adj. *p*-value < 0.05, absolute log_2_ fold change > 2; green = no significant difference, orange = significant difference). (**C**) Representative pictures of hiPS-HEP derived from ChiPSC12 (**C1**, **C4**), ChiPSC18 (**C2**, **C5**), and ChiPSC22 (**C3**, **C6**) on day 12 post-thawing after three hours incubation with LDL-DyLight (**C1**–**C3**) or untreated control cells (**C4**–C**6**). Scale bars 50 µm. Abbreviation: LDL: low-density lipoprotein.

**Figure 5 ijms-21-00469-f005:**
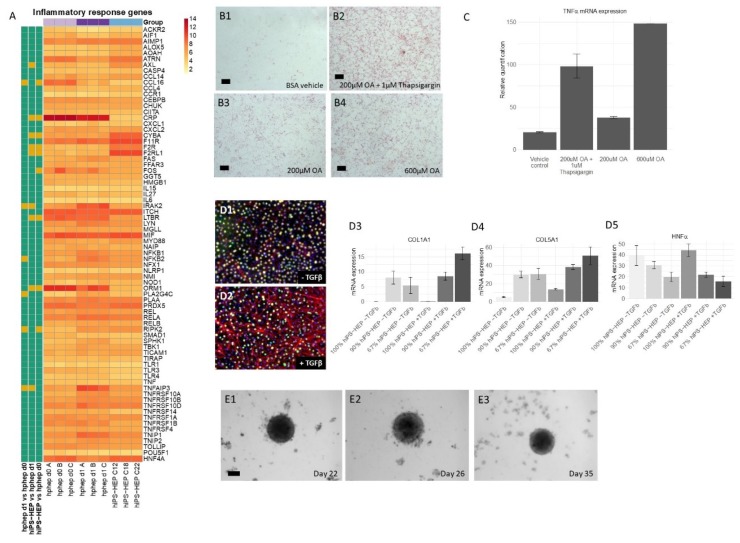
Inflammatory response in hiPS-HEP and co-cultures of hiPS-HEP and primary hepatic stellate cells (HSC) for non-alcoholic steatohepatitis (NASH) modeling. (**A**) Heatmap of mRNA expression of genes involved in the inflammatory response in hiPS-HEP derived from three hiPSC lines (ChiPSC12, ChiPSC18, ChiPSC22) on day 13 post-thawing and hphep directly after thawing (d0) and on day 1 post-thawing (d1). The panel to the left of the heatmap indicates significant differences between the groups (adj. *p*-value < 0.05, absolute log_2_ fold change > 2; green = no significant difference, orange = significant difference). (**B**) Representative stainings of hiPS-HEP on day 6 post-thaw derived from ChiPSC18 with Oil Red O visualizing lipid accumulations after a 24 h incubation with bovine serum album (BSA) vehicle (**B1**), 200 µM OA and 1 µM Thapsigargin (**B2**), 200 µM OA (**B3**), and 600 µM OA (**B4**), respectively. Scale bars 100 µm. (**C**) *TNFα* mRNA expression in hiPS-HEP on day 6 post-thaw derived from ChiPSC18 after a 24 h incubation with BSA vehicle, 200 µM OA and 1 µM Thapsigargin, 200 µM OA, and 600 µM OA, respectively. (**D1**–**D2**) Representative pictures of HNF4α (green) and α-SMA (red) immunostainings of co-cultures of hiPS-HEP derived from ChiPSC18 and hepatic stellate cells (HSC) without (**D1**) and with (**D2**) a 4 day treatment with 20 ng/mL TGFβ, respectively. Nuclear counterstaining with DAPI (blue). (**D3**–**D5**) *Collagen 1A1* (**D3**), *Collagen 5A1* (**D4**), and *HNF4α* (**D5**) mRNA expression in cultures consisting of 100% hiPS-HEP: 0% HSC, 90% hiPS-HEP: 10% HSC, or 67% hiPS-HEP: 33% HSC with and without a 4 day treatment with 20 ng/mL TGFβ, respectively. Error bars represent standard deviation of technical quadruplicates. (**E**) Representative pictures of 3D spheroids consisting of hiPS-HEP and HSC (ratio 2:1) cultured for 22 days (**E1**), 26 days (**E2**), and 35 days (**E3**), respectively. Scale bar 100 µm. Abbreviations: α-SMA: α-smooth muscle actin; DAPI: 4′,6-diamidino-2-phenylindole; HNF4α: Hepatocyte Nuclear Factor 4α; HSC: hepatic stellate cells; OA: Oleic acid; TGFβ: transforming growth factor β; TNFα: Tumor necrosis factor α.

**Figure 6 ijms-21-00469-f006:**
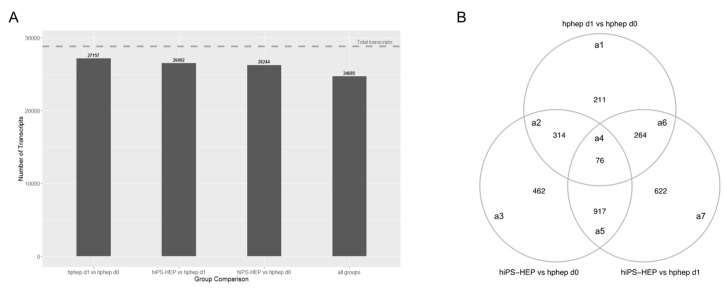
Global gene expression analysis of transcriptomics data. (**A**) Number of similarly expressed mRNA transcripts for the different group comparisons. (**B**) Venn diagram of 2866 differentially expressed genes (DEGs) between the different group comparisons: hphep d0 vs. hphep d1 (area a1), hphep d0 vs. hiPS-HEP (area a3), hphep d1 vs. hiPS-HEP (area a7), hphep d0 vs. hphep d1 and hphep d0 vs. hiPS-HEP (area a2), hphep d0 vs. hiPS-HEP and hphep d1 vs. hiPS-HEP (area a5), hphep d0 vs. hphep d1 and hphep d1 vs. hiPS-HEP (area a6), and all three groups (area a4).

**Table 1 ijms-21-00469-t001:** Donor demographics of the human primary hepatocytes.

Donor	Age	Gender	Ethnicity	Cause of Death	Viability
**KFF**	56	male	Caucasian	anoxia, 2nd to cardiac arrest	95%
**YEM**	46	female	Caucasian	intracerebral hemorrhage—stroke	87%
**MSW**	69	male	African-American	intracerebral hemorrhage, 2nd to cerebrovascular accident (stroke)	91%

**Table 2 ijms-21-00469-t002:** CYP activity substrate cocktail.

CYP	Substrate	Assay Concentration
**1A**	Phenacetin	10 μM
**2B6**	Bupropion	10 μM
**2C19**	Mephenytoin	50 μM
**2C9**	Diclofenac	10 μM
**2D6**	Bufuralol	10 μM
**3A**	Midazolam	5 μM

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
