# Peer review of "Characterization of Human Induced Pluripotent Stem Cell-Derived Hepatocytes with Mature Features and Potential for Modeling Metabolic Diseases"

_ijms, 2020, doi:10.3390/ijms21020469_

Round 1
Reviewer 1 Report
Comment:
Manuscript ID: ijms-654414, by Gustav Holmgren, Benjamin Ulfenborg *, Annika Asplund, Karin Toet, Christian X Andersson, Ann Hammarstedt, Roeland Hanemaaijer, Barbara Küppers-Munther, Jane Synnergren, entitled “Characterization Of Human Pluripotent Stem Cell-Derived Hepatocytes With Adult Features And Potential For Modeling Metabolic Diseases, ” is of great interest. The logical flow and coherence of narratives should be improved by addressing the following 21 specific comments.
Specific Comments:
The title: “Characterization of Human Pluripotent Stem Cell-Derived Hepatocytes. With Adult Features and Potential for Modeling Metabolic Diseases” is read odd, as “Adult Features” are not well defined.
Lines 18-19: “human pluripotent stem cell-derived hepatocytes (hiPS-HEP)” misled the reader to iPSC. Line 72: “Human pluripotent stem cells (hiPSC).” Can they clearly define? PSC can be derived from ESC and iPSC. So, which one did they work?
Line 21: “The transcriptomics analysis revealed that 86% of the genes” – the percentage is not critical, but the critical panel of the gene is. E.g., Line 426: “HNF4α and found that it was expressed in 92% of hiPS-HEP cells” vs. Line 443: “the homogenous HNF4α-expression in hiPS-HEP,” – if they got 8% of the cells didn’t express HNF4α, how did they define the homogeneity? In reality, heterogeneity is the nature of any stem cells.
Lines 29-30: “hiPS-HEP could be co-cultured with primary hepatic stellate cells” – why is it unique?
Line 30: “for modeling NASH” – spell out NASH.
Line 38: “over 500 different functions in the body” – how did they define such a list? Where is their reference?
Line 52: “the rapid loss of their functionality in culture” – by what assays?
Line 72: “inexhaustible source of cells” – misleading, as nothing is inexhaustible.
Line 95: “2.1. Homogeneous population of hiPSC-derived hepatocytes with mature hepatocyte features” – what’s their definition of Homogeneous population? In reality, single-cell gene expression profiling shows a huge variation in one batch of cells in culture (Oncotarget. 2018 Sep 7;9(70):33290-33301. doi: 10.18632/oncotarget.26044). Line 110: “subpopulation in both hiPS-HEP” – by what standard?
Lines 111-112: “The immunostaining of these three markers was very similar for hiPS-HEP derived from all three hiPSC lines (Suppl. Figure 1).” How did they define “similar?” Without quantity, how did they determine such?
1A: “three hiPSC lines (ChiPSC12, ChiPSC18, ChiPSC22)” – it appears they differed from hphep cells. Why lower? How many batches of populations did they do? Even more pronounced in Fig. 1D & E – why? Due to the “low expression of ARG1 in hiPS-HEP compared to hphep d1?” If they claimed these cells are similar, why such huge differences?
Table 1. Donor demographics of the human primary hepatocytes. How did they integrate into the logic flow?
2A, B, C: – it appears they differed from hphep cells of these biomarkers. Why?
3C “Glycogen deposition” – any quantification?
Lines 241-251: “essential feature of metabolically functional hepatocytes is the physiological response” – Why didn’t they assay glucose uptake? Gene expression is a real-time physiological response.
5. Line 339: “Representative pictures of 3D spheroids consisting of hiPS-HEP and HSC (ratio 2:1) cultured” – Did they have any QC control for spheroids? The size? The structure? The gene expression profile? The functional assays?
Lines 438-442: “Besides high homogeneity, the reproducibility of the differentiation protocol is of great importance so that different cell batches show the same characteristics. We found that different hiPS HEP batches derived from the same hiPSC line showed little overall variation; see for example expression of hepatocyte markers (Figure 1A), expression of the drug-metabolizing machinery (Figure 2A-C), CYP activity results (Figure 2D) and insulin signaling (Figure 3D).” Neither “high homogeneity,” nor “the same characteristics” did they offer the relevant definition in a quantified fashion. Alternatively, any physiological function assays?
Lines 453-455 “Albumin secretion, which is one hallmark for adult hepatocytes, was on similar levels in hiPS-HEP and hphep (Figure 1D), which was confirmed by similar Albumin mRNA levels in both cell types (Figure 1A).” –Albumin secretion was not similar – nor “similar Albumin mRNA levels” could be confirmed by their data.
Lines 573-576: “Our results highlight the great promise of hiPSC-derived hepatocytes as an unlimited cell source to build more advanced cell models, to better recapitulate the human liver environment, to reflect the population diversity and to increase the predictability for disease modeling.” The statement overstated beyond their data: Neither the human liver environment, nor the population diversity, nor the predictability for disease modeling, were weighed in with their data sets.
Lines 579-581: “Cryopreserved hiPS-HEPs derived from the hiPSC lines ChiPSC12, ChiPSC18, and ChiPSC22 were thawed, plated, and maintained according to the vendor instructions (Cellartis Enhanced hiPS581 HEP v2 kits, Cat.no. Y10133, Y10134, Y10135, Takara Bio, takarabio.com).” Could they give any peer-reviewed publications on these cell lines?
Author Response
Reviewer 1
Comments and Suggestions for Authors
Comment:
Manuscript ID: ijms-654414, by Gustav Holmgren, Benjamin Ulfenborg *, Annika Asplund, Karin Toet, Christian X Andersson, Ann Hammarstedt, Roeland Hanemaaijer, Barbara Küppers-Munther, Jane Synnergren, entitled “Characterization Of Human Pluripotent Stem Cell-Derived Hepatocytes With Adult Features And Potential For Modeling Metabolic Diseases, ” is of great interest. The logical flow and coherence of narratives should be improved by addressing the following 21 specific comments.
Specific Comments:
The title: “Characterization of Human Pluripotent Stem Cell-Derived Hepatocytes. With Adult Features and Potential for Modeling Metabolic Diseases” is read odd, as “Adult Features” are not well defined.
Response: We agree that the title can be improved, and it has been changed to “Characterization of Human Induced Pluripotent Stem Cell-Derived Hepatocytes With Mature Features and Potential for Modeling Metabolic Diseases”
Lines 18-19: “human pluripotent stem cell-derived hepatocytes (hiPS-HEP)” misled the reader to iPSC. Line 72: “Human pluripotent stem cells (hiPSC).” Can they clearly define? PSC can be derived from ESC and iPSC. So, which one did they work?
Response: We thank the reviewer for pointing out this ambiguity. In this study only iPSC-derived cells have been used and the terminology is adjusted throughout the whole manuscript to clearly state that.
Line 21: “The transcriptomics analysis revealed that 86% of the genes” – the percentage is not critical, but the critical panel of the gene is. E.g., Line 426: “HNF4α and found that it was expressed in 92% of hiPS-HEP cells” vs. Line 443: “the homogenous HNF4α-expression in hiPS-HEP,” – if they got 8% of the cells didn’t express HNF4α, how did they define the homogeneity? In reality, heterogeneity is the nature of any stem cells.
Response: We agree with the reviewer that the word homogeneity should be used with caution. To avoid ambiguity in the results the text has been rephrased at several places in the manuscript to clarify that the cell cultures are not 100% homogeneous with respect to HNF4α expression. Row 448-451 now reads:
”To evaluate the hepatocyte content in the hiPS-HEP cultures, we stained for the key hepatocyte transcription factor HNF4α and found that it was expressed in 92% of hiPS-HEP cells, thus indicating a highly efficient differentiation result and a near-homogeneous culture regarding HNF4α expression”.
And row 468-470 now reads:
”In contrast to the nearly uniform HNF4α-expression in hiPS-HEP, other markers or features are expressed only in a subpopulation of both hiPS-HEP and hphep, e.g., Albumin, α1-Antitrypsin, ASGPR1, and glycogen storage (Figure 1)”.
Lines 29-30: “hiPS-HEP could be co-cultured with primary hepatic stellate cells” – why is it unique?
Response: To the best of our knowledge, this is the first time this has been achieved. We discuss this in line 566-572 in the discussions part and have also added one sentence in the introduction to clarify this point.
Line 87-88: “Furthermore, primary hepatic stellate cells could be activated in 2D co-cultures with hiPS-HEP which, to the best of our knowledge, is the first time that this is reported.”
Line 30: “for modeling NASH” – spell out NASH.
Response: We thank the reviewer for the careful review of the manuscript. NASH is now spelled out at first use.
Line 31: ”non-alcoholic steatohepatitis (NASH)”
Line 38: “over 500 different functions in the body” – how did they define such a list? Where is their reference?
Response: We have added a reference in line 40 and changed the text accordingly.
Line 39-40: “The liver is an organ that spans a large variety of different functions in the body: energy metabolism, detoxification, and production of serum proteins and bile, just to mention a few [1].”
Line 52: “the rapid loss of their functionality in culture” – by what assays?
Response: Typical functional assays for characterization of hphep are drug metabolizing assays, which now are added to the text including relevant references. Line 52-54 now reads:
“The shortage of relevant human donor material, the large donor variation, and the rapid loss of their functionality, e.g. of the expression of the drug metabolizing machinery in culture [2-5], are the most prominent problems.”
Line 72: “inexhaustible source of cells” – misleading, as nothing is inexhaustible.
Response: We agree with the reviewer, this phrasing is misleading. The text is rephrased, and line 73-75 now reads.
“Human induced pluripotent stem cells (hiPSC) are a virtually unlimited source of cells and have the potential to differentiate into specialized cell types, which provides unique opportunities for usage in a wide range of applications.”
Line 95: “2.1. Homogeneous population of hiPSC-derived hepatocytes with mature hepatocyte features” – what’s their definition of Homogeneous population? In reality, single-cell gene expression profiling shows a huge variation in one batch of cells in culture (Oncotarget. 2018 Sep 7;9(70):33290-33301. doi: 10.18632/oncotarget.26044).
Response: The reviewer has a good point in asking for a more defined phrasing and is right when pointing out variations within populations. As we discuss at several points in the manuscript, hepatocytes are heterogeneous by nature with regard to many functions, the so-called metabolic zonation. Therefore, we used HNF4α, a general hepatocyte marker expressed in all hepatocytes regardless of their position in the liver lobe, in order to assess the differentiation efficiency by measuring the hepatocyte content in the hiPS-HEP cultures. And with regard to the HNF4α-immunostaining, the hiPS-HEP cultures were found to be highly homogeneous.
Line 110: “subpopulation in both hiPS-HEP” – by what standard?
Response: These subpopulations became apparent by immunostainings for ASGPR1, AAT and Albumin. This has been clarified in the manuscript on line 119-123, which now reads:
“In contrast to the highly uniform HNF4α expression, ASGPR1, α1-Antitrypsin (AAT), and Albumin (Alb) appeared to be more strongly expressed in a subpopulation in both hiPS-HEP and hphep cultures (Figure 1C3-C5 and 1C7-9, respectively). The immunostaining of these three markers was comparable for hiPS-HEP derived from all three hiPSC lines in terms of staining intensity (Suppl. Figure 1).”
Lines 111-112: “The immunostaining of these three markers was very similar for hiPS-HEP derived from all three hiPSC lines (Suppl. Figure 1).” How did they define “similar?” Without quantity, how did they determine such?
Response: We agree with the reviewer that the use of ‘similar’ should be used with care and put in a context. In this sentence similar refers to staining intensity through ocular assessment. The sentence is rephrased, and line 121-123 now reads:
“The immunostaining of these three markers was comparable for hiPS-HEP derived from all three hiPSC lines in terms of staining intensity (Suppl. Figure 1).”
1A: “three hiPSC lines (ChiPSC12, ChiPSC18, ChiPSC22)” – it appears they differed from hphep cells. Why lower? How many batches of populations did they do? Even more pronounced in Fig. 1D & E – why? Due to the “low expression of ARG1 in hiPS-HEP compared to hphep d1?” If they claimed these cells are similar, why such huge differences?
Response: The reviewer is correct, the hiPS-HEP differ from hphep in some functions and markers whereas others are on a comparable level in both cell types. We point this out and discuss these differences at several places in the manuscript. The reason that some functions are not optimal yet in hiPS-HEP and require further improvement is that it is a challenge to achieve full functionality for a cell type which has so many different functions. In order to evaluate how similar these two cell types are, we have performed a global analysis of the transcriptomics data which showed that 86% of transcripts were similarly expressed in hiPS-HEP and hphep with a CV<10%. It is based on this global analysis that we make the conclusion that the cells are similar.
For clarification, we have added the number of batches in the figure 1 and 2 legends (lines 145 and 212), in addition to stating this in the material and methods section (line 640).
Table 1. Donor demographics of the human primary hepatocytes. How did they integrate into the logic flow?
Response: We agree with the reviewer that the donor demographics table is more logic to place in section 4.1 and it is now moved to line 636.
2A, B, C: – it appears they differed from hphep cells of these biomarkers. Why?
Response: The reviewer is right that hiPS-HEP differ in expression of some phase I and II enzymes as well as some transporters from hphep. Possible explanations for the lower expression levels could be for example: 1) incomplete maturity of the hiPS-HEP, or 2) an effect of the 2D culture conditions as a downregulation of these genes is also observed in hphep cultured in 2D.
3C “Glycogen deposition” – any quantification?
Response: We chose in this study to determine the glycogen storage by PAS staining in order to see if subpopulations of cells were PAS-positive, which is characteristic for the liver, instead of using a quantification method, which gives an average value for all cells.
Lines 241-251: “essential feature of metabolically functional hepatocytes is the physiological response” – Why didn’t they assay glucose uptake? Gene expression is a real-time physiological response.
Response: We thank the reviewer for the suggestion of additional interesting analyses, and it would indeed be exciting to also look at the glucose uptake as well as other functional characteristics of the liver. We have now added the glucose transporter GLUT2 (SLC2A2), which is involved in glucose uptake in the liver, to Figure 3A (line 289). However, functional assays for glucose uptake are not in the scope for the present work but will be considered in future studies.
Line 339: “Representative pictures of 3D spheroids consisting of hiPS-HEP and HSC (ratio 2:1) cultured” – Did they have any QC control for spheroids? The size? The structure? The gene expression profile? The functional assays?
Response: These are merely initial results from a proof-of-concept study without a detailed characterization. A thorough characterization will be the scope of future studies. We noticed that we did not add a scale bar in Figure 5E1 (line 338) and have added one now. We apologize for missing this.
Lines 438-442: “Besides high homogeneity, the reproducibility of the differentiation protocol is of great importance so that different cell batches show the same characteristics. We found that different hiPS HEP batches derived from the same hiPSC line showed little overall variation; see for example expression of hepatocyte markers (Figure 1A), expression of the drug-metabolizing machinery (Figure 2A-C), CYP activity results (Figure 2D) and insulin signaling (Figure 3D).” Neither “high homogeneity,” nor “the same characteristics” did they offer the relevant definition in a quantified fashion. Alternatively, any physiological function assays?
Response: The paragraph has been rephrased according to the reviewer’s comment and quantitative measure in terms of coefficient of variation (CV) has been added. The paragraph at line 463-467 now reads:
We found that different hiPS-HEP batches derived from the same hiPSC line showed overall low variation measured as coefficient of variation (CV). For example, expression of hepatocyte markers (Figure 1A) show CV<5% for all genes except for ASGR1(CV<8%), and expression of the drug metabolizing machinery show CV<15% for the majority (87%) of all genes in Figure 2A-C.
Lines 453-455 “Albumin secretion, which is one hallmark for adult hepatocytes, was on similar levels in hiPS-HEP and hphep (Figure 1D), which was confirmed by similar Albumin mRNA levels in both cell types (Figure 1A).” –Albumin secretion was not similar – nor “similar Albumin mRNA levels” could be confirmed by their data.
Response: We thank the reviewer for pointing out a misleading wording at this point. This sentence at line 479-483 has been rephrased with quantitative measurements and it now reads:
Albumin secretion, which is one hallmark for adult hepatocytes, was measured at comparable levels in hiPS-HEP derived from line C12 and hphep (CV<9%; Figure 1D) and with slightly higher secretion in the hiPS-HEP derived from line C18 and C22. This was confirmed by comparable Albumin mRNA levels across both cell types (CV<9%; Figure 1A).
Lines 573-576: “Our results highlight the great promise of hiPSC-derived hepatocytes as an unlimited cell source to build more advanced cell models, to better recapitulate the human liver environment, to reflect the population diversity and to increase the predictability for disease modeling.” The statement overstated beyond their data: Neither the human liver environment, nor the population diversity, nor the predictability for disease modeling, were weighed in with their data sets.
Response: We agree with the reviewer that this statement was too visionary and has been rephrased to describe what will be required to build more advanced cell models. Line 601-606 now reads:
Our results highlight the potential of hiPSC-derived hepatocytes as a virtually unlimited cell source. In order to build more advanced cell models, to better recapitulate the human liver environment, to reflect the population diversity and to increase the predictability for disease modeling, advanced 3D culture systems including multiple cell types and microfluidics as well as panels of hiPSC lines derived from various ethnicities need to be developed and implemented.
Lines 579-581: “Cryopreserved hiPS-HEPs derived from the hiPSC lines ChiPSC12, ChiPSC18, and ChiPSC22 were thawed, plated, and maintained according to the vendor instructions (Cellartis Enhanced hiPS581 HEP v2 kits, Cat.no. Y10133, Y10134, Y10135, Takara Bio, takarabio.com).” Could they give any peer-reviewed publications on these cell lines?
Response: To the best of our knowledge, there are no publications available yet where the hiPS-HEP have been used. However, these lines are commercially available products with tech notes available online with characterization data (https://www.takarabio.com/products/stem-cell-research/stem-cells-and-stem-cell-derived-cells/human-stem-cell-derived-hepatocytes).
References
References mentioned in the responses to reviewers are listed here. They are numbered differently in the manuscript.
Trefts, E., M. Gannon, and D.H. Wasserman, The liver. Curr Biol, 2017. 27(21): p. R1147-r1151.
Guguen-Guillouzo, C. and A. Guillouzo, General review on in vitro hepatocyte models and their applications. Methods Mol Biol, 2010. 640: p. 1-40.
Rodriguez-Antona, C., et al., Cytochrome P450 expression in human hepatocytes and hepatoma cell lines: molecular mechanisms that determine lower expression in cultured cells. Xenobiotica, 2002. 32(6): p. 505-20.
Ulvestad, M., et al., OATP1B1/1B3 activity in plated primary human hepatocytes over time in culture. Biochem Pharmacol, 2011. 82(9): p. 1219-26.
Ulvestad, M., et al., Drug metabolizing enzyme and transporter protein profiles of hepatocytes derived from human embryonic and induced pluripotent stem cells. Biochem Pharmacol, 2013. 86(5): p. 691-702.
Cicchini, C., et al., Snail controls differentiation of hepatocytes by repressing HNF4alpha expression. J Cell Physiol, 2006. 209(1): p. 230-8.
Lu, H., Crosstalk of HNF4alpha with extracellular and intracellular signaling pathways in the regulation of hepatic metabolism of drugs and lipids. Acta Pharm Sin B, 2016. 6(5): p. 393-408.
Reviewer 2 Report
The paper of Holmgren et al. deals with description of functional potential of pluripotent stem cell- derived hepatocytes and demonstrates their ability to model hepatic diseases. This in depth analysis of metabolic potential has investigated the main hepatic characteristics important to be known when evaluating the quality of the cells regarding human primary hepatocytes. Many informative data are provided, highly relevant for the reader, particularly those drawn from the transcriptomic analysis. However, the paper suffers from several shortcomings and the manuscript has to be improved before being suitable for publication in IJMS.
-Description of fundamental characteristics of the cells is missing: a picture of good quality showing general structural organization of confluent cells with evidence of polarized bile canaliculi, is requested; this reviewer is also questioning about karyotype of the cells and its variability; is-it the same from one batch to the other? Are the cells arrested at confluence when differentiated? Thus, a short general description of the model will be useful for the reader.
-localization of transporters in G figure 2 is not informative and has to be improved; CDFDA transport to biliary pole would be better relevant test
-glycogen localization in figure 3: C1 and C2 show hepatocytes, however, cells from C3 have not or have lost hepatocyte morphology, suggesting heterogeneous population of bad hepatic morphology ; primary hepatocytes in C4 are also of very poor quality. Figure showing intracellular accumulation of lipid has to be improved or to be deleted
-experiments with TGFbeta are very interesting for demonstrating stellate cell activation; however, conclusion is not completely clear regarding HNF4 in Figure 5, TGFbeta being reported to partly inhibit HNF4, particularly in cells originating from stem cells. Discussion on this point should be added.
- the stemness origin of the cells leads to question on the possible expression of stem cell markers whih could modify their behavior in conditions mimicking diseases; this point has be onsidered and briefly discussed in the discussion section.
Introduction and discussion do not mention any references on other cell lines derived from stem-cells or from differentiated tumor cell lines; in total which benefits do we get in using the hipPS-hep cells instead of others?.
Author Response
Reviewer 2
Comments and Suggestions for Authors
The paper of Holmgren et al. deals with description of functional potential of pluripotent stem cell- derived hepatocytes and demonstrates their ability to model hepatic diseases. This in depth analysis of metabolic potential has investigated the main hepatic characteristics important to be known when evaluating the quality of the cells regarding human primary hepatocytes. Many informative data are provided, highly relevant for the reader, particularly those drawn from the transcriptomic analysis. However, the paper suffers from several shortcomings and the manuscript has to be improved before being suitable for publication in IJMS.
-Description of fundamental characteristics of the cells is missing: a picture of good quality showing general structural organization of confluent cells with evidence of polarized bile canaliculi, is requested; this reviewer is also questioning about karyotype of the cells and its variability; is-it the same from one batch to the other? Are the cells arrested at confluence when differentiated? Thus, a short general description of the model will be useful for the reader.
Response: We thank the reviewer for pointing out that a fundamental description of the cells was missing and this has been added, Line 102-106 now reads:
“The hiPS-HEP cultures consisted of large cells with a polygonal cell shape and a dark cytoplasm (Figure 1C1), which is typical for hepatocytes. HiPS-HEP were plated at confluency after thawing and did not proliferate, indicating terminal differentiation. Importantly, expression of stemness markers such as Oct4 (POUF51) were as low in hiPS-HEP as in hphep (see heatmap in Figure 1B), which indicates the loss of the stemness features as a result of an efficient hepatocyte differentiation.”
A representative morphology picture has also been added in Figure 1C1 (line 137). The banks of hiPSCs are routinely karyotyped as part of the production process at Takara Bio, which certifies that the hiPSCs do not carry any chromosomal aberrations. However, the hepatocytes derived from the hiPSC are not karyotyped.
-localization of transporters in G figure 2 is not informative and has to be improved; CDFDA transport to biliary pole would be better relevant test
Response: The reviewer is right in pointing out that the immunostainings in Figure 2G do not show the formation of bile canaliculi. They only confirm the expression of the transporters on the protein level. We have performed CDFDA assays but could not observe the formation of bile canaliculi. We have added this information in the manuscript. Line 199-202 now reads:
“The protein expression of MRP2, NTCP, BSEP, and OATP1B1 were confirmed by immunostainings (Figure 2G1-4), however, formation of bile canaliculi was not observed when performing a CDFDA staining (data not shown) indicating that no cell polarity is formed in hiPS-HEP cultures.”
-glycogen localization in figure 3: C1 and C2 show hepatocytes, however, cells from C3 have not or have lost hepatocyte morphology, suggesting heterogeneous population of bad hepatic morphology ; primary hepatocytes in C4 are also of very poor quality. Figure showing intracellular accumulation of lipid has to be improved or to be deleted
Response: We agree with the reviewer that there is little Glycogen staining in Figure 3C3 compared to C1 and C2. We have seen this repeatedly for this particular hiPSC line and it would be interesting to investigate why this line has less Glycogen deposits than the other lines. For visualizing the Glycogen staining, we did not use phase contrast settings in the microscope and without phase contrast it is difficult to see the cell morphology and one may get the impression of poor morphology. We have added a representative phase contrast picture in Figure 1C1. The reviewer is right that the hphep are a bit sparse, but the staining is nevertheless informative. In our experience, hphep generally have little glycogen deposits after freezing/thawing which is why we cultured them for 24hr before staining for glycogen.
Regarding the lipid staining pictures, we again did not use phase contrast settings in the microscope since this interfered with seeing the Oil Red Staining. In our humble opinion, the important information in these pictures is the amount of Oil Red O staining and this is clearly visible. We hope that the reviewer agrees with this.
-experiments with TGFbeta are very interesting for demonstrating stellate cell activation; however, conclusion is not completely clear regarding HNF4 in Figure 5, TGFbeta being reported to partly inhibit HNF4, particularly in cells originating from stem cells. Discussion on this point should be added.
Response: We thank the reviewer for pointing this out. We have added two new references and several sentences for discussing this. Line 375-383 now reads:
Since TGFβ has been reported to suppress HNF4α expression [6] and HNF4α expression is crucial for maintaining a differentiated hepatocyte phenotype [7], we investigated HNF4α expression both on mRNA and protein level in the co-cultures. In general, HNF4α mRNA expression was lower in co-cultures than in hiPS-HEP monocultures independent from TGFβ-treatment which was expected and can be explained with a decreased hiPS-HEP content in the co-cultures due to the addition of HSC (Figure 5D5). Importantly, TGFβ-treatment of the co-cultures tended to decrease HNF4α mRNA expression levels (Figure 5D5) but the observed differences were not significant. In accordance with this, fewer HNF4α-immune-positive nuclei could be observed the TGFβ-treated cultures (compare Figure 5D1 and 5D2).
-the stemness origin of the cells leads to question on the possible expression of stem cell markers which could modify their behavior in conditions mimicking diseases; this point has been considered and briefly discussed in the discussion section.
Response: The reviewer has a good point in questioning the expression of stemness markers in the hiPS-HEP. We have included the stemness marker Oct4 (POUF5F1) in all heatmaps in order to mark background signals but have failed to mention or explain this in the text. We thank the reviewer for bringing this to our attention and have now added a sentence describing these results in line 104-106:
“Importantly, expression of stemness markers such as Oct4 (POUF51) is as low in hiPS-HEP as in hphep (see heatmap in Figure 1B) which indicates the loss of the stemness features as a result of an efficient hepatocyte differentiation.”
-Introduction and discussion do not mention any references on other cell lines derived from stem-cells or from differentiated tumor cell lines; in total which benefits do we get in using the hipPS-hep cells instead of others?.
Response: For this study, we have chosen hphep for benchmarking since they are considered the gold standard. Many hepatoma cell lines do not recapitulate important functional aspects of hepatocytes and are therefore generally not considered to be useful or relevant for benchmarking. However, we agree with the reviewer that it would be interesting to compare to multiple control cell types but this is unfortunately beyond the scope of the present study which is why we picked the one cell type regarded to be most relevant.
Regarding hepatocytes derived from human pluripotent stem cells by other researchers, we do refer to a selection of relevant publications in both introduction and discussion. Since the field and the number of publications is rapidly growing it is not possible to refer to all relevant publications in the scope of a research paper.
References
References mentioned in the responses to reviewers are listed here. They are numbered differently in the manuscript.
Trefts, E., M. Gannon, and D.H. Wasserman, The liver. Curr Biol, 2017. 27(21): p. R1147-r1151.
Guguen-Guillouzo, C. and A. Guillouzo, General review on in vitro hepatocyte models and their applications. Methods Mol Biol, 2010. 640: p. 1-40.
Rodriguez-Antona, C., et al., Cytochrome P450 expression in human hepatocytes and hepatoma cell lines: molecular mechanisms that determine lower expression in cultured cells. Xenobiotica, 2002. 32(6): p. 505-20.
Ulvestad, M., et al., OATP1B1/1B3 activity in plated primary human hepatocytes over time in culture. Biochem Pharmacol, 2011. 82(9): p. 1219-26.
Ulvestad, M., et al., Drug metabolizing enzyme and transporter protein profiles of hepatocytes derived from human embryonic and induced pluripotent stem cells. Biochem Pharmacol, 2013. 86(5): p. 691-702.
Cicchini, C., et al., Snail controls differentiation of hepatocytes by repressing HNF4alpha expression. J Cell Physiol, 2006. 209(1): p. 230-8.
Lu, H., Crosstalk of HNF4alpha with extracellular and intracellular signaling pathways in the regulation of hepatic metabolism of drugs and lipids. Acta Pharm Sin B, 2016. 6(5): p. 393-408.
Round 2
Reviewer 2 Report
Thanks for improvements of the manuscript. Since you show that the cell lines fail to get polarity, meaning that intracellular trafficking of bile acids or any drugs (clearance as well) and other metabolites are not corrected addressed to biliary or plasma domains, several applications regarding diseases will be poorly relevant, it would be good if the authors could limit their conclusions in term of possible applications of the model.
Author Response
Comments from reviewer 2 (minor)
Thanks for improvements of the manuscript. Since you show that the cell lines fail to get polarity, meaning that intracellular trafficking of bile acids or any drugs (clearance as well) and other metabolites are not corrected addressed to biliary or plasma domains, several applications regarding diseases will be poorly relevant, it would be good if the authors could limit their conclusions in term of possible applications of the model.
Response:
We agree with the reviewer that the limitation in polarity should be clearly pointed out in the manuscript and this has now been added in the discussion at line 506-509, which now reads:
”In addition, formation of bile canaliculi was not observed, meaning that hiPS-HEP do not develop cell polarity with apical and basolateral membrane compartments. This limits the utility of the cells for applications requiring bile canaliculi formation, e.g., biliary excretion of drugs.